Resource

# High-throughput phenomics of global ant biodiversity

**A list of authors and their affiliations appears at the end of the paper**

The big data era in biology is underway, but the study of organismal form has been slow to capitalize on advances in imaging and computation. Imaging approaches can digitize whole organisms, but low throughput has limited the effort to document morphological diversity. Here, within the open science initiative 'Antscan', we applied high-throughput synchrotron X-ray microtomography to capture phenotypes across a diverse and ecologically dominant insect group: ants. At https://www.antscan.info, we provide 2,193 whole-body three-dimensional ant datasets from 212 genera and 792 species to broadly cover the ant phylogeny with a global scope, also pairing phenomic data with genome sequencing projects. Scans acquired with standardized parameters facilitate automated analysis, and free access to data can broaden the audience and incentivize methods development. Antscan presents a scalable approach to create libraries of diverse anatomies, heralding an era of studies on the evolution, structure and function of organismal phenotypes.

The diversity of life is manifested in the endless forms of organisms. Grasping this phenotypic variation is one key to understanding the evolution of organismal diversity, the interface between genomic variation and the environment, the engineering principles in nature, the functional traits relevant to ecosystem processes and the responses of organisms to global changes[1–3].

Scientific collections are of fundamental value for documenting life's diversity through space and time and form the foundation for basic and applied biodiversity science. To unlock this value, genetic sequencing, photography and photogrammetry are established approaches for generating digital information from standing insect collections[4–7]. Truly capturing organismal form, the digitization of internal and external three-dimensional (3D) morphological data offers the potential to expand the use of collections even further: for example, to combine morphological and anatomical traits of form with molecular phylogenies and correlate evolutionary patterns with ecological data[8,9]. However, big data approaches to morphology and anatomy lag behind rapid advances in other digitization efforts.

X-ray computed microtomography (micro-CT) can non-destructively digitize valuable collection material in 3D[10,11], but substantial challenges must be overcome to achieve a phenomic big data revolution across the millions of animal species[9,10,12–14]. Among the major groups of animals, insects are the most abundant and species-rich class. They exhibit a vast diversity of body plans and morphological adaptations to environmental and social challenges. Insects play key functions in terrestrial ecosystems, are critical for agriculture, serve as inspiration for biomimetic design and are important indicator organisms for measuring the effects of global change. Toward the goal of documenting morphological diversity, synchronizing multiple facilities with conventional laboratory micro-CT scanners is one way forward, and this model has been successful for scanning vertebrates[15]. However, to image insects with conventional micro-CT, staining and/or drying is often required to discriminate between tissues[16–19], which not only necessitates additional preprocessing steps but alters specimens and excludes some collection material from imaging. For the high magnification required to image smaller organisms, limits in photon flux density (the amount of radiation emitted from the X-ray source) result in long scan times, easily exceeding several hours or more. Therefore, the bottlenecks in preprocessing and throughput call for an alternative approach to achieve digitization milestones for soft and hard tissue diversity across the global insect fauna.

Further, toward the broader utilization of micro-CT data in biology, data availability represents a formidable challenge because datasets are often not openly available after publication[8] and tedious processing of 3D data is still an impediment to analyses[20]. Automation using computer-vision and artificial intelligence methods can greatly reduce manual input[21,22], which necessitates highly comparable data for training and application. Yet, in laboratory micro-CT, image properties

✉e-mail: julian.katzke@oist.jp; fhitagarcia@gmail.com; economo@umd.edu; thomas.vandekamp@kit.edu

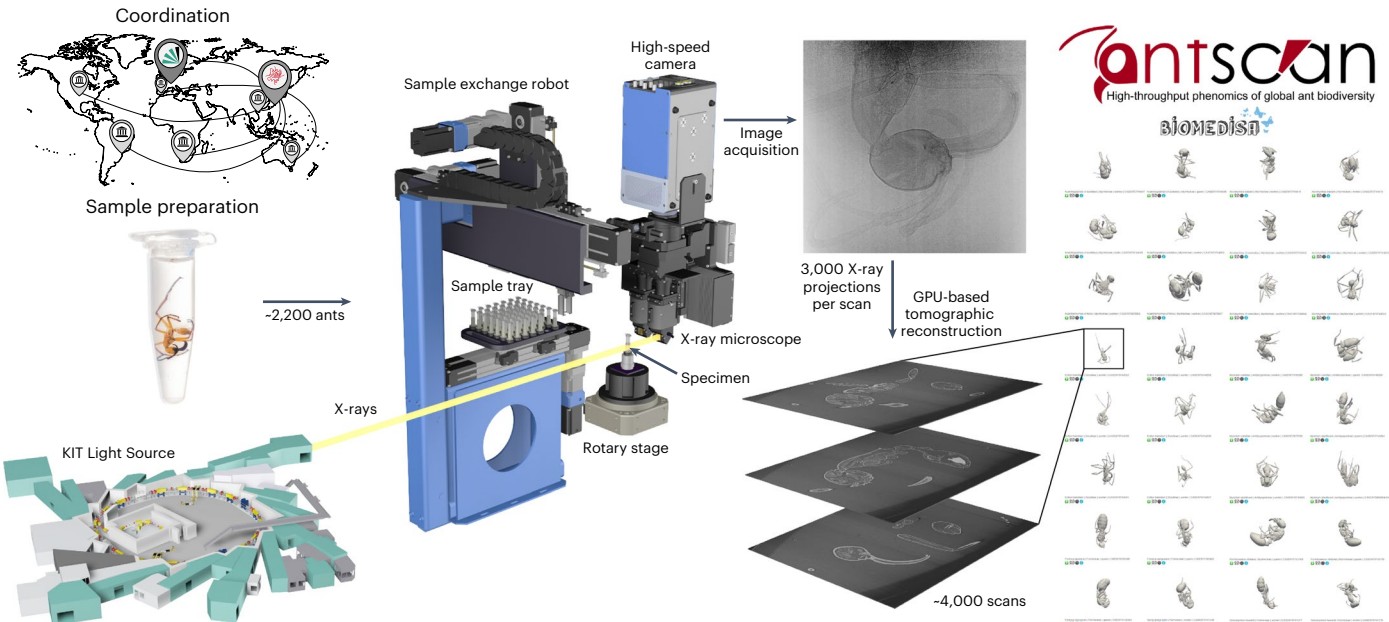

**Fig. 1 | Overview of the Antscan workflow.** Left: specimens were centralized and prepared before high-throughput synchrotron micro-CT scanning at KIT Light Source. Middle: high-throughput synchrotron micro-CT employs the synchrotron beam as X-ray source, a sample exchange robot, a rotary stage, a high-resolution detector system and a high-speed camera. Image stacks are then computed from the projections using GPU-based tomographic reconstruction. Right: the scans are made publicly available in the online database for download, signified here by a snapshot of 3D previews.

differ largely among specimens as staining levels and scanning conditions vary[23], which makes applying such methods more difficult. Thus, large-scale 3D digitization efforts must also address issues of image comparability and open access.

Synchrotron-based micro-CT offers the potential to perform standardized 3D imaging of large numbers of ethanol-preserved collection samples without additional invasive preparation. It features high flux density, allowing short scanning times at high resolution, and broad contrast capabilities, particularly phase contrast, enabling the visualization of soft tissue without staining. These advantages can be optimally exploited by automated high-throughput setups with robotic sample exchangers[24]. Automated synchrotron micro-CT has already been successful to digitize invertebrates for comparative research[24,25] and is suitable to be extended to cover entire groups of diverse small organisms.

Within the 'Antscan' initiative, we created a massive open resource for research on the phenotypic diversity of ants (Fig. 1). Ants occupy many ecological niches that coincide with enormous morphological and anatomical variation between and within species. They live in colonies, most incorporating worker and queen female castes and males. Many species also exhibit distinct subcastes or more continuous variation among workers. Systematics based on robust molecular phylogenies[26–33] and extensive ecological and behavioral research[34,35] will enable contextualizing morphology in ecological and evolutionary research. As a globally distributed, morphologically variable and ecologically dominant group of insects, ants form an excellent basis for a pilot initiative in digitizing insect biodiversity in 3D.

We gathered vouchered ant samples from museums and personal collections worldwide. As soft-tissue preservation is an important issue, we exclusively selected ethanol-preserved specimens. The ants were scanned in toto (whole-body) within ethanol using a high-throughput synchrotron micro-CT setup. Standardized scanning and 3D reconstruction protocols facilitated the creation of comparable datasets and, consequently, the application of machine learning and computer-vision approaches. The processed datasets provide the foundation for the Antscan repositories, accessible via https://www.antscan.info, which offer free access to the entire 3D image data collection of the initiative. As we synchronized our efforts with large-scale genome sequencing projects, such as the Global Ant Genomics Alliance[36,37], by targeting conspecific ants or even scanning ants from the same nest series sampled for those projects, we established a vital connection between molecular and morphological data, enhancing the scientific value of both datasets. We highlight the possibility of creating complex digital 3D models by semi-automatic segmentation and show how simple screening through a vast number of datasets can quickly recover patterns in morphological and ecological adaptations.

Antscan serves as a vast open resource for promoting research on the morphology and anatomy of ants but also establishes a powerful workflow that can be adapted for other lineages of small organisms across the tree of life. By digitizing and democratizing an important aspect of scientific collections while strictly keeping the data tied to their sources and collectors (Supplementary Table 1), these valuable resources become accessible for analysis worldwide. Making the exceptional phenotypic diversity of ants available in 3D thus unlocks a previously inaccessible wealth of information for a broader audience, enabling comprehensive analyses and contributing to the establishment of online repositories for researchers, educators or nature enthusiasts.

## Results

### Phylogenetic coverage

We designed specimen sampling within Antscan to be phylogenetically broad, to represent species-poor clades, to include rarely collected species and to sample multiple representatives of highly diverse genera. Datasets cover 14 out of the 16 currently recognized extant subfamilies, and ants were identified to 212 out of 343 presently recognized genera[30,38] (Fig. 2). These 212 genera incorporate more than 90% of all described ant species. Out of 2,193 scanned ants, 1,711 specimens were identified to 659 species; and within 482 not fully identified ant specimens, we currently recognize at least 133 distinct morphospecies, resulting in at least 792 ant species represented in our total dataset. Antscan includes 1,671 workers, 291 queens and 220 males. Within the worker caste, we identified 18 ants as explicitly being media workers and 125 as major workers, with more polymorphic ants yet to be delimited.

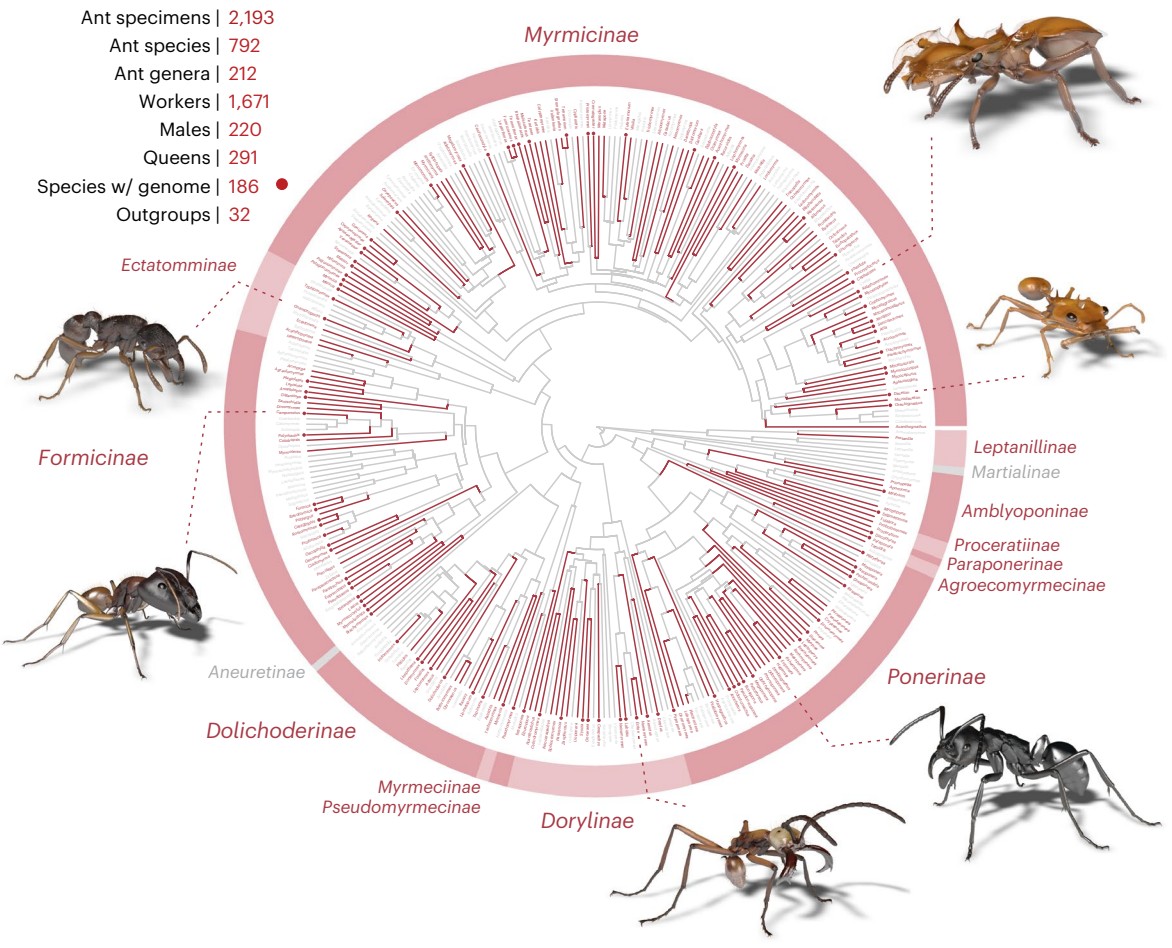

Ant specimens | 2,193
Ant species | 792
Ant genera | 212
Workers | 1,671
Males | 220
Queens | 291
Species w/ genome | 186 ●
Outgroups | 32

**Fig. 2 | The diverse set of species and genera across the ant tree of life covered by Antscan.** The ant phylogeny with genera sampled in Antscan displayed in red and missing genera in gray. A dot at the end of a genus branch indicates that at least one genome from a species within that genus is sequenced. Examples of 3D models derived from the micro-CT scans shown clockwise from the top: *Cephalotes* sp., *Daceton armigerum*, *Streblognathus peetersi*, *Eciton hamatum*, *Camponotus* sp., *Gnamptogenys* aff. *continua*. The phylogenetic tree is based on Economo et al.[33] with missing genera added through taxonomic affinities or other published phylogenies[26–32].

In addition to the ant datasets, we provide 32 scans of non-ant wasps for outgroup comparison. For the three most species-rich ant genera *Camponotus*, *Pheidole* and *Strumigenys*, which incorporate more than 850 species each, we sampled 37, 33 and 39 species, respectively. Important currently monotypic genera like *Apomyrma*, *Santschiella* and *Tatuidris* are included; variations of peculiar morphologies, such as various trap-jaw ants and multiple turtle ant species, are covered; and globally important species like fire ants and Argentine ants were sampled, allowing for the investigation of a plethora of research questions building on the global diversity of ants.

In coordination with genome sequencing projects, 186 scanned species are associated with genomic data. These are represented by 585 individual ant scans, of which 157 were taken from the same nest series as the sequenced specimens. As many taxa have further been covered by published molecular phylogenies, with Antscan, we aim to form a platform for future comparative research on the genomic basis of phenotypic variation and diversification in invertebrates.

## Tomographic data

By employing high-throughput synchrotron micro-CT at two beamlines of the KIT Light Source, we generated in toto tomograms for 2,193 ant plus 32 outgroup specimens. Three magnification settings with corresponding fields of view were employed to optimally cover the wide range of ant sizes in the collection (Fig. 3). Specimens exceeding the field of view vertically were scanned in several height steps.

In addition to absorption contrast, synchrotron measurements also exploit phase contrast due to the partial transverse coherence of the radiation. Phase contrast arises during the free-space propagation of the transmitted wave field, which already after short propagation distances enhances the tissue boundaries in the measured images, so-called edge enhancement[39]. This is particularly important for soft tissues such as muscles, which absorb almost no radiation. A phase-retrieval algorithm then converts the edge enhancement into distinguishable tissue contrast. To obtain good overall contrast for both higher-absorbing parts like the exoskeleton and soft tissues, the primary datasets of the Antscan database are blended volumes, composed of one volume obtained by standard 3D reconstruction of the measured images and the other by additional phase retrieval applied before the 3D reconstruction.

We used a modified imaging pipeline for a limited number of specimens. We included 132 ants that were originally prepared for a laboratory micro-CT project[22]. These specimens were stained with iodine to enhance soft-tissue contrast. Due to their high X-ray absorption, phase retrieval was not applicable in these cases. The diameter of the six largest specimens exceeded even the largest field of view available at the synchrotron setup. We therefore stained them with iodine and scanned them with a laboratory microtomograph. Although the iodine-stained specimens deviated from the standardized imaging protocol, we were able to incorporate the resulting tomograms into the downstream processing steps.

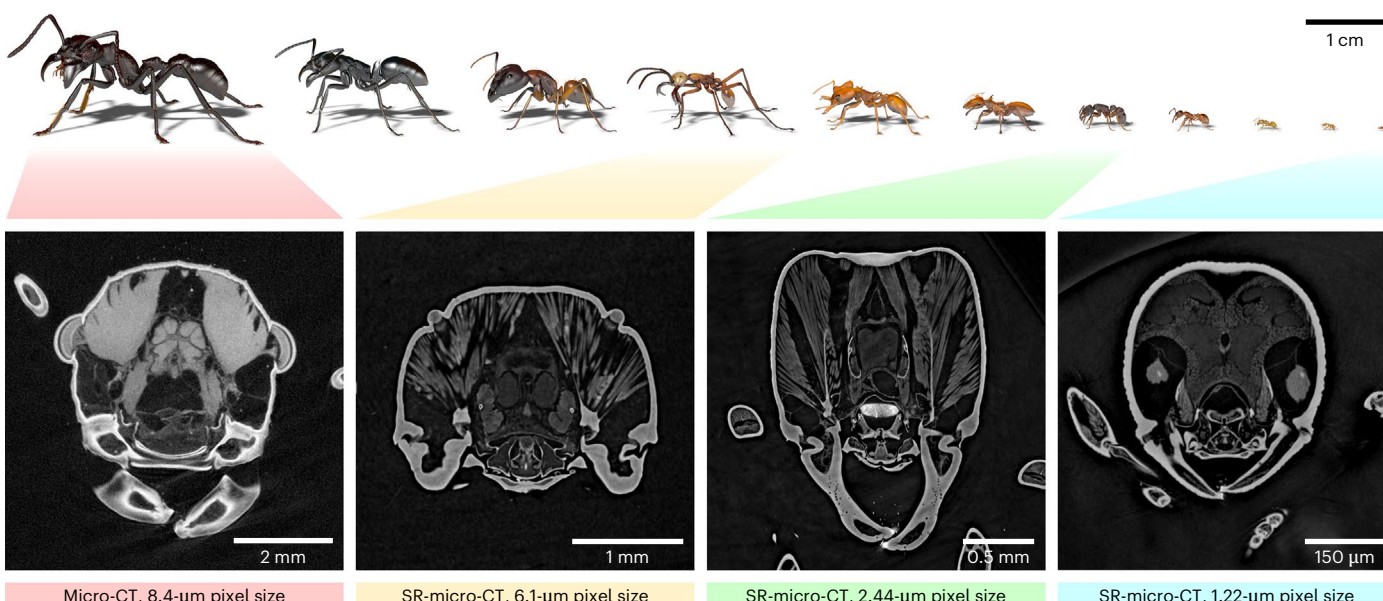

**Fig. 3 | Sample images showing different magnifications for ant workers of different sizes.** Top: 3D models of various ant workers scanned within Antscan depicted to illustrate operational scale. Bottom: image slices from the heads of four ants acquired with different magnifications. From left to right: *Paraponera clavata* stained with iodine and scanned using laboratory micro-CT as one of the largest ants, which did not fit inside the largest field of view of the synchrotron setup; *Eciton hamatum* subsoldier as a large ant; *Gnamptogenys* aff. *continua* as a medium-sized ant; *Discothyrea sexarticulata* as one of the smallest ants. SR, synchrotron radiation.

Although the original tomograms are saved as 32-bit data, we converted and further processed the tomograms as 8-bit tagged image file format (TIFF) stacks. When required, we used a computer-vision workflow to merge individual height steps automatically into a combined volumetric dataset, showing the entire ant body. Employing a neural network using Biomedisa[21,40], we further crudely segmented datasets automatically and cropped all tomograms by removing background and thus reducing file size considerably. This automated processing of all data underlines the suitability of high-throughput synchrotron micro-CT for future large-scale analyses using computer vision and machine learning.

Because the data acquisition and processing parameters were identical for the sample subsets scanned at a given magnification at a beamline, the reconstructed gray values of the corresponding tissues within these subsets are equivalent (except for few iodine-treated samples), ensuring comparability and facilitating the application of machine learning and computer-vision approaches.

For quality assurance, we visually inspected all scans to check for specimen preservation. Most ants were well preserved, but as is unavoidable when drawing from standing collections, some showed severe damage from soft-tissue shrinkage and decay. This was probably caused by DNA extraction, exposure to air during transport, handling or storage. However, as the rigid exoskeleton generally was not affected, many morphological or morphometric analyses can still be applied. Overall, given the preservation quality, we conclude that both short- and long-term stored ethanol-preserved material is generally suitable for nondestructive high-throughput synchrotron micro-CT to digitally preserve 3D anatomy of invertebrates indefinitely.

Because samples were prepared and positioned manually beforehand and scanning proceeded automatically, the partial truncation of some datasets could not be avoided. Legs and antennae of larger specimens within a magnification category were particularly prone to be outside the field of view during scanning. Future developments toward improved robotic setups and imaging pipelines that can autonomously recognize the individual position of specimens during measurements and readjust scanning positions accordingly will likely resolve such minor issues.

## The public Antscan repositories

The public Antscan database infrastructure is curated, aims to provide an interactive experience and is sustainable. All tomograms acquired within Antscan, 3D surface models and associated metadata are provided publicly under CC BY 4.0 license in open repositories that can be accessed via the Antscan website (https://www.antscan.info). The interface for the primary, interactive repository (https://biomedisa.info/antscan) was created on the image analysis and segmentation platform Biomedisa[40].

In the interactive repository, Antscan data are easily accessible, can be visually evaluated before use and can be directly processed. A search function allows users to search the database for specific keywords in the metadata. Individual datasets are presented on the home page with previews of 3D surface models based on automatic neural network segmentation. The ant datasets then provide additional previews for image slices and the surface mesh as an interactive 3D model. On the navigation page, download buttons for each specimen provide direct access to the tomograms. Alternatively, each scan can be accessed for more detailed information and additional data downloads. In addition to 3D data, each ant is accompanied by extensive metadata, including taxonomic rank, ecological parameters and unique specimen identifiers (Supplementary Table 1). We also provide information on specimen locality, additional geographical details, habitat and whether a sequenced genome is associated with the specimen. The metadata also provide the information necessary to identify and credit the curators, other contributors and host institutions for each specimen. Integration into the Biomedisa interface allows registered users to perform semi-automatic and automatic volumetric image segmentation with the Biomedisa application, and processed datasets can be easily shared with other Biomedisa users.

In addition, we ensure the sustainability of the entire Antscan database by providing another long-term secure location for all processed image files and metadata at KIT's RADAR4KIT repository (https://radar.kit.edu/radar/en/search?query=antscan). We associated a digital object identifier (DOI) with each scanned ant specimen to provide a permanent digital signature linking ants and scans, which is also noted in the metadata (Supplementary Table 1). Using this system, unique

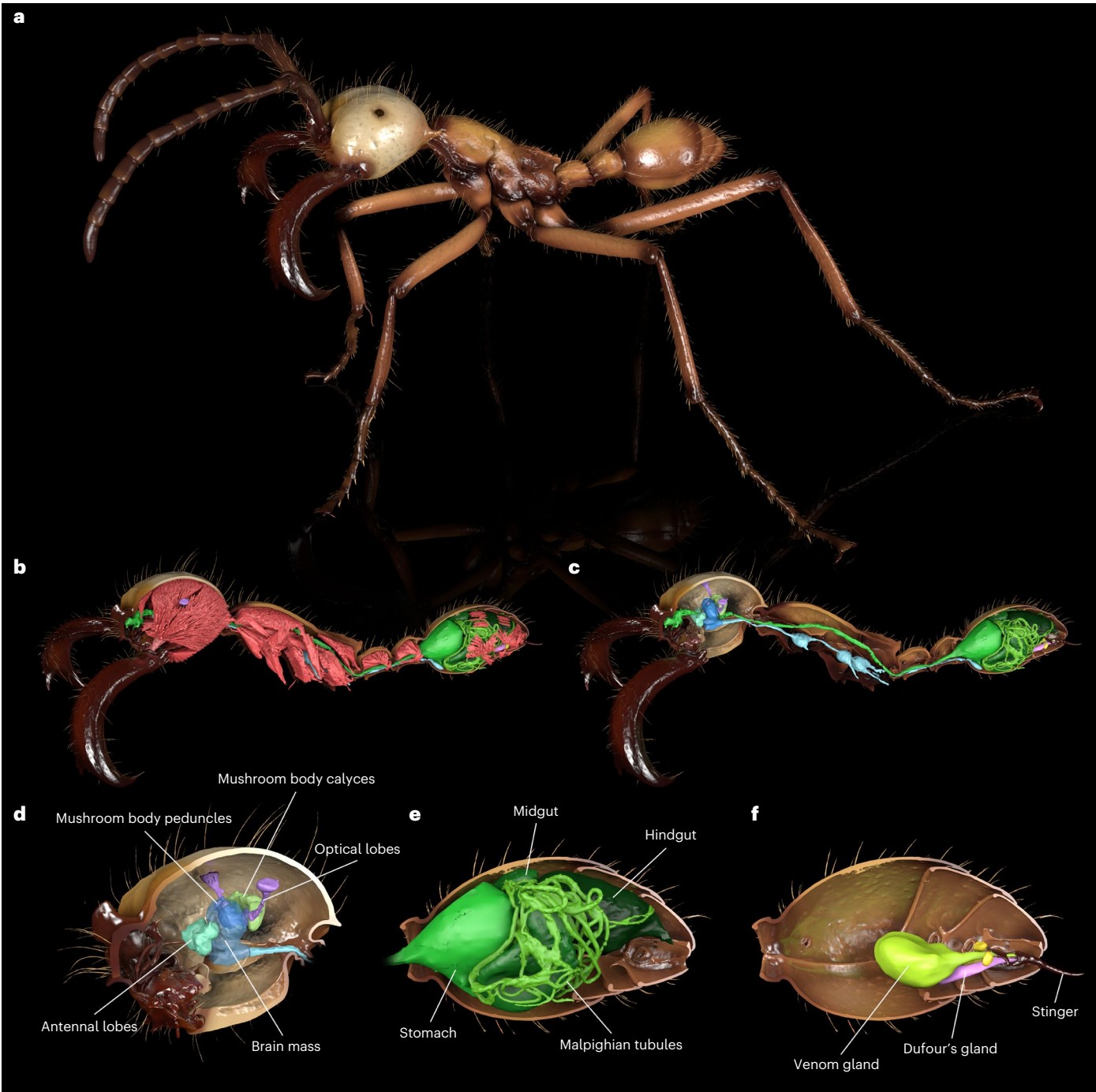

**Fig. 4 | Renderings of an exemplary Antscan specimen (*Eciton hamatum* subsoldier CASENT0744582).** Renderings show the segmented cuticle and tissues representative of the level of detail captured with synchrotron micro-CT. **a**, Full habitus of the ant with an animated, more life-like pose and colors inspired by photographs. **b**, Cuticle cut at the sagittal section revealing internal tissues with muscles in red occupying most of the internal space in an ant's body. **c**, Removing the muscles reveals the digestive tract (green) and the nervous system (blue). **d**–**f**, Zoomed-in renderings focusing on the ant brain (**d**), gut (**e**) and sting apparatus (**f**), respectively.

IDs exist for both the physical specimens and the generated scans. The chosen format of a mirrored, open database allows for smooth phylogeny/taxonomy-based online navigation and ensures long-term public availability of the Antscan data.

## Exemplary use cases

To demonstrate the wealth of information contained in an individual ant scan, we segmented the exoskeletal elements, muscles, nervous tissues, sting apparatus and digestive tract of a subsoldier of the South American army ant *Eciton hamatum* (Fig. 4). Such segmented anatomical features can, for example, be extracted and quantified for analyses, visualized as renderings (Figs. 2, 3 and 4a), animated (Supplementary Video 1) and 3D-printed.

Moreover, the vast number of available specimen data facilitate large-scale comparative studies: for example, to identify similarities and differences between lineages and to trace the evolution of traits throughout the ant tree of life. In this context, not all scientific cases require extensive segmentations of individual ants. Simple screening

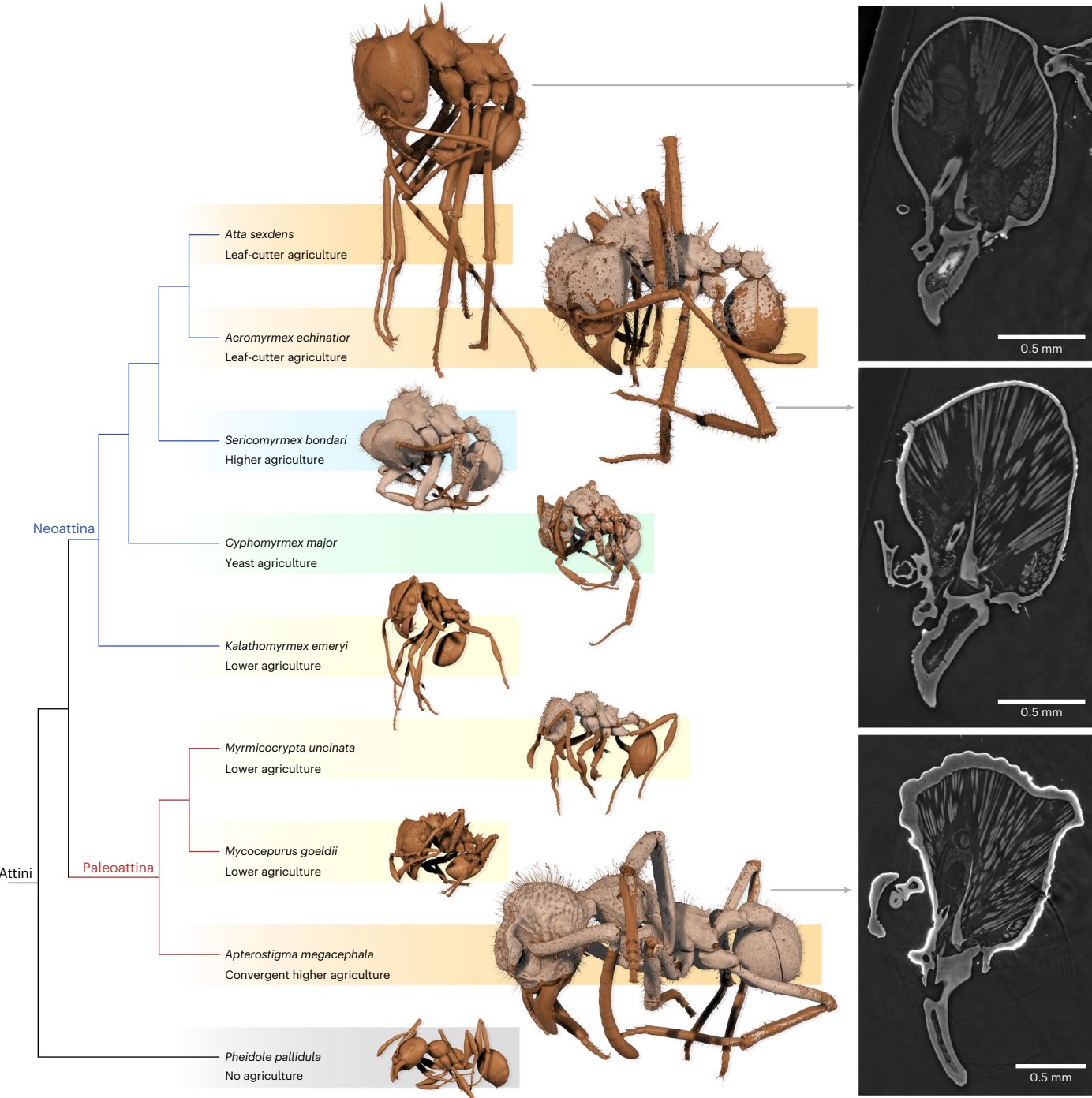

**Fig. 5 | Trait recognition with Antscan on the example of biomineral armor.** Comparative screening revealed that biomineral armor is very common among fungus-growing ants and was found in several attine species. Right, the slices through the tomograms show the armor as a thin but distinct coat of highly absorbing (brighter) material on top of the cuticle. Left, in the 3D renderings it is visualized as a beige color and the cuticle is displayed as a darker brown. The cladogram is based on the phylogeny by Hanisch et al.[43], with gradient colors behind taxon names and 3D models indicating the different degrees of fungus agriculture.

of the datasets may quickly clarify whether morphological features are present or absent in ant species or entire lineages. Here this is shown on the example of biomineral armor, an unusual insect trait first described for *Acromyrmex echinatior*[41] and previously unknown in other ants. Antscan data immediately revealed that this trait is more widely distributed. A conspicuous highly absorbing layer on the surface of the cuticle confirms that biomineral armor is in fact common among fungus-growing ants (Myrmicinae: Attini) and scattered across the different agricultural systems that evolved in these clades[42,43] (Fig. 5). Within the attine ants, it appears to be absent in the genera *Atta*, *Kalathomyrmex* and *Mycocepurus*, which is consistent with an inferred secondary loss of biomineral armor[41]. We have not found evidence of this trait in any species outside the Attini. This example illustrates the potential for standardized high-throughput imaging datasets to enable testing evolutionary and ecological hypotheses at larger scales without the acquisition of new data.

## Discussion

With Antscan, we provide an open resource that spans the morphological and anatomical diversity of ants. The scope of the Antscan database

necessitated a scalable design to meet the demands for large-scale 3D digitization of phenotypes from biodiversity collections. Our database is open, and the metadata identifies all contributors[8,11,44].

High-throughput synchrotron micro-CT is a key technology to effectively overcome the bottleneck of acquisition times in micro-CT. It is fast and provides well-resolved morphological datasets with high contrast for both hard and soft tissues if specimens have been properly preserved. Including file transfer, which currently constitutes a technological bottleneck in our scanning setup, Antscan moved at a pace of ~25 scans per hour. Similarly resolved scans of ants using conventional micro-CT would take about 12 hours each on average[45,46], with the additional delay of individual scan setups. Including larger ants scanned in height steps, extrapolating this to the 4,010 individual scans performed, it would have taken one laboratory micro-CT scanner operated around the clock more than six years to obtain a similar dataset.

To ensure the taxonomic breadth, image quality and usability of the Antscan datasets, it was necessary to design a coordinated pipeline around the high-throughput imaging experiment. International collaboration among collection stakeholders included managing institutional and governmental requirements but translates into scientific value through accurate documentation of specimens and accountability for contributors. The process of sorting, shipping and preparing specimens to fit the robotic setup is laborious but essential for achieving optimal scans. Reconstructing tomographic volumes from X-ray projections is technically demanding and computationally intensive. Antscan raw data and public database storage demands exceed 200 terabytes. These challenges highlight the need for continued investment in computing and storage infrastructure and workflow automation. In addition, there is an urgent need to ease data processing[20,47] and to use computer-vision methods to reliably distinguish materials in micro-CT data[21,22]. The automated segmentation methods demonstrated here for cropping scans and generating 3D surface models demonstrate the potential for Antscan datasets to address these issues. Together, the pipeline and the resulting database provide a foundational resource for advancing comparative morphology in the digital era.

High-throughput synchrotron micro-CT in close collaboration with collectors and collection managers promises a key solution for digitizing 3D anatomy across small invertebrates. One remaining obstacle to the broader use of synchrotron tomography is the limited beamtime availability at synchrotron facilities. Most of them offer user service based on peer-reviewed proposals, and obtaining beamtime remains competitive. Automation of synchrotron facility imaging stations through robots and the resulting high-throughput setups are also developed to very different degrees. Moreover, the homogeneously illuminated field of view at many synchrotron X-ray imaging beamlines is restricted, which impedes the scanning of larger specimens. However, current developments, such as longer beamlines[48] and Bragg crystal optics[49], and sample exchange systems point toward a broader range of organism and collection sizes to be imaged with comparable synchrotron micro-CT setups.

Massive, largely standardized resources of morphological biodiversity such as Antscan have the potential to be transformative for integrative biological research. The scientific community benefits from a rapidly growing collection of commercial and open-access software that facilitates analysis and visualization of 3D volumetric datasets and a willingness to publish workflow recommendations and best practices[50,51]. Volume renderings based on gray values can be employed to quickly generate impressive 3D views[52,53], and surface meshes allow digital dissections and the creation of interactive 3D models[45,54]. Segmentation remains the most common feature-extraction method for CT scans[51], and the standardized contrast properties across Antscan datasets facilitate segmentation techniques. Based on neural network segmentation, we already provide simple 3D surface mesh models for all specimens, which allow examination of external morphology and morphometric measurements. As Antscan data include external and internal anatomy across diverse social insects, with the added potential to integrate 3D anatomy with genomic information[36], Antscan supports a wide range of integrative use cases, including morphometric, biomechanical and physiological studies. Analyzing cuticle versus body volumes for 507 species from Antscan, Matte et al.[55] found that thinner cuticle is associated with larger colonies and higher diversification rates. With more than 2,000 high-resolution micro-CT datasets, Antscan enables larger scales of comparison for evolutionary and morphological research than previously possible with tomographic data, especially for insects.

In contrast to physical specimens, digital information can be directly accessed from anywhere in the world, enabling immediate and simultaneous access by researchers, artists and the public, thus allowing wider and more equitable engagement with biodiversity. Antscan is a scalable approach to digitize small-bodied organisms in 3D to make the 'micro' world more accessible. Like a genome, a 3D scan contains deep information about an organism, but obtaining quantitative information from micro-CT scans remains challenging. We will need new bioinformatics based on automated image analysis to fully unlock the potential of databases like Antscan, but recent developments in this area show promise[21,22]. Antscan aims to empower and encourage people around the world to engage with and incorporate highly resolved ant morphology and anatomy into their science, education and art. With the approach described here and further developments in imaging technology, bioinformatics and artificial intelligence on the horizon, it is time for the study of phenotypes to take its place alongside other big data endeavors in biology.

## Online content

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

Julian Katzke ®[1,52] ✉, Francisco Hita Garcia ®[1,2,52] ✉, Philipp D. Lösel ®[3], Fumika Azuma[1], Tomáš Faragó ®[4,5], Lazzat Aibekova[1], Alexandre Casadei-Ferreira ®[1], Shubham Gautam[1,6], Adrian Richter[1], Evropi Toulkeridou ®[1], Sabine Bremer[5], Elias Hamann ®[4], Jenny Hein[4], Janes Odar[5], Chandan Sarkar[5], Marcus Zuber ®[4], Jacobus J. Boomsma[7], Rodrigo M. Feitosa[8], Lukas Schrader ®[9], Guojie Zhang[10,11], Sándor Csősz[12,13], Minsoo Dong ®[1], Olivia Evangelista[14], Georg Fischer[1], Brian L. Fisher ®[15], Jaime A. Florez-Fernandez ®[14], The Antscan GAGA Consortium*, Fede García[16], Kiko Gómez ®[16], Donato A. Grasso ®[17], Stephane de Greef[18], Benoit Guénard[19], Peter G. Hawkes ®[20,21], Robert A. Johnson[22], Roberto A. Keller ®[23], Rasmus S. Larsen[11], Timothy A. Linksvayer ®[22], Cong Liu ®[24], Arthur Matte ®[1,25], Masako Ogasawara[26], Hao Ran ®[27], Juanita Rodriguez ®[14], Enrico Schifani ®[28], Ted R. Schultz ®[29], Jonathan Z. Shik[11], Jeffrey Sosa-Calvo ®[29,30], Chao Tong ®[22], Leonardo Tozetto[1], Seonwoo Yoon ®[31], Masashi Yoshimura[26], Jie Zhao[32], Tilo Baumbach[4,5], Evan P. Economo ®[1,33,52] ✉ & Thomas van de Kamp ®[4,5,52] ✉

[1]Biodiversity and Biocomplexity Unit, Okinawa Institute of Science and Technology Graduate University, Onna-son, Japan. [2]Center for Integrative Biodiversity Discovery, Museum für Naturkunde, Berlin, Germany. [3]Department of Materials Physics, Research School of Physics, The Australian National University, Canberra, Australian Capital Territory, Australia. [4]Institute for Photon Science and Synchrotron Radiation (IPS), Karlsruhe Institute of Technology (KIT), Eggenstein-Leopoldshafen, Germany. [5]Laboratory for Applications of Synchrotron Radiation (LAS), Karlsruhe Institute of Technology (KIT), Karlsruhe, Germany. [6]Bullab, Department of Ecology, Faculty of Environmental Sciences, Czech University of Life Sciences Prague, Prague, Czech Republic. [7]Section for Ecology and Evolution, Department of Biology, University of Copenhagen, Copenhagen, Denmark. [8]Departamento de Zoologia, Universidade Federal do Paraná, Curitiba, Brazil. [9]Institute for Evolution & Biodiversity, University of Münster, Münster, Germany. [10]Center of Evolutionary & Organismal Biology, Zhejiang University School of Medicine, Hangzhou, China. [11]Department of Biology, University of Copenhagen, Copenhagen, Denmark. [12]HUN-REN-ELTE-MTM Integrative Ecology Research Group, Biological Institute, Eötvös Loránd University, Budapest, Hungary. [13]Department of Systematic Zoology and Ecology, Institute of Biology, ELTE-Eötvös Loránd University, Budapest, Hungary. [14]Australian National Insect Collection, CSIRO, Canberra, Australian Capital Territory, Australia. [15]Department of Entomology, California Academy of Sciences, San Francisco, CA, USA. [16]Independent Researcher, Barcelona, Spain. [17]Insect Ethology, Ecology and Sociobiology Lab, Department of Chemistry, Life Sciences and Environmental Sustainability, University of Parma, Parma, Italy. [18]Independent researcher, Brussels, Belgium. [19]School of Biological Sciences, The University of Hong Kong, Hong Kong, Hong Kong, China. [20]AfriBugs CC, Pretoria, South Africa. [21]SARChI Chair on Biodiversity Value and Change in the Vhembe Biosphere Reserve, University of Venda, Thohoyandou, South Africa. [22]School of Life Sciences, Life Sciences Center, Arizona State University, Tempe, AZ, USA. [23]Centre for Ecology, Evolution and Environmental Changes & CHANGE - Global Change and Sustainability Institute, Museu Nacional de História Natural e da Ciência, Universidade de Lisboa, Lisbon, Portugal. [24]Department of Organismic and Evolutionary Biology, Museum of Comparative Zoology, Harvard University, Cambridge, MA, USA. [25]Department of Zoology, University of Cambridge, Cambridge, UK. [26]Environmental Science and Informatics Section, Okinawa Institute of Science and Technology Graduate University, Onna-son, Japan. [27]Biological Education and Research Laboratory, Mancheng High School of Hebei Province, Baoding, China. [28]Institute of Evolutionary Biology, CSIC - University Pompeu Fabra, Barcelona, Spain. [29]Department of Entomology, National Museum of Natural History, Smithsonian Institution, Washington, DC, USA. [30]Department of Biology, University of Central Florida, Orlando, FL, USA. [31]Urban Agriculture Division, Ulsan City Agricultural Technology Center, Ulsan, South Korea. [32]State Key Laboratory of Genetic Resources and Evolution, Kunming Institute of Zoology, Chinese Academy of Sciences, Kunming, China. [33]Department of Entomology, University of Maryland, College Park, MD, USA. [52]These authors contributed equally: Julian Katzke, Francisco Hita Garcia, Evan P. Economo, Thomas van de Kamp. *A list of authors and their affiliations appears at the end of the paper. ✉e-mail: julian.katzke@oist.jp; fhitagarcia@gmail.com; economo@umd.edu; thomas.vandekamp@kit.edu

**The Antscan GAGA Consortium**

**Serge Aron**[34], **Abel Bernadou**[35], **Martin Bollazzi**[36], **Raphaël Boulay**[37,53], **Sylvia Cremer**[38], **Heike Feldhaar**[39], **Susanne Foitzik**[40], **Erik T. Frank**[41], **Jürgen Gadau**[9], **Daniele Giannetti**[17], **Stephane de Greef**[18], **Heikki Helanterä**[42], **Ana Ješovnik**[43], **Fredrick Larabee**[44], **Bálint Markó**[45], **David Nash**[7], **Jérôme Orivel**[46], **Jes Søe Pedersen**[7], **Frédéric Petitclerc**[46], **Stephen Rehner**[47], **Morten Schiøtt**[48], **András Tartally**[49], **Kazuki Tsuji**[50], **Irene Villalta**[37] & **Herbert C. Wagner**[51]

[34]Evolutionary Biology & Ecology, Université Libre de Bruxelles, Brussels, Belgium. [35]Centre de Recherches sur la Cognition Animale, Centre de Biologie Intégrative, Université de Toulouse, CNRS, UPS, Toulouse, France. [36]Entomología. Facultad de Agronomía, Universidad de la República, Montevideo, Uruguay. [37]Research Institute on Insect Biology (IRBI), CNRS, Université de Tours, Tours, France. [38]ISTA (Institute of Science and Technology Austria), Klosterneuburg, Austria. [39]Animal Population Ecology, Bayreuth Center of Ecology and Environmental Research (BayCEER), University of Bayreuth, Bayreuth, Germany. [40]Institute of Organismic and Molecular Evolution, Johannes Gutenberg University Mainz, Biozentrum, Mainz, Germany. [41]Department of Animal Ecology and Tropical Biology, Biocenter, University of Würzburg, Würzburg, Germany. [42]Ecology and Genetic Research Unit, University of Oulu, Oulu, Finland. [43]Croatian Myrmecological Society, Zagreb, Croatia. [44]Department of Biological Sciences, San Jose State University, San Jose, CA, USA. [45]Hungarian Dept. of Biology and Ecology, Babes-Bolyai University, Cluj-Napoca, Romania. [46]UMR EcoFoG (AgroParisTech, CIRAD, CNRS, INRAE, Université des Antilles, Université de Guyane), Campus Agronomique, Kourou, France. [47]USDA-ARS, Mycology and Nematology Genetic Diversity and Biology Laboratory, Beltsville, MD, USA. [48]Department of Bioengineering, Technical University of Denmark, Lyngby, Denmark. [49]Department of Evolutionary Zoology and Human Biology, University of Debrecen, Debrecen, Hungary. [50]Department of Agro-Environmental Sciences, University of the Ryukyus, Okinawa, Japan. [51]Institute of Biology, University of Graz, Graz, Austria. [53]Deceased: Raphaël Boulay.

## Methods

### Material

We gathered ants preserved in ethanol from museum, university and private collections. One hundred thirty-two specimens were previously stained with iodine (Supplementary Table 1). At the Okinawa Institute of Science and Technology (OIST), we checked the vials, chose apparently well-preserved specimens, transferred them to fresh 99% ethanol, assigned unique specimen IDs and, depending on their size, stored ants individually in 0.2-ml, 0.5-ml and 1.5-ml plastic reaction tubes that would fit in the robotic setup at KIT. We extracted and databased label data from the specimens to link relevant ecological metadata to the specimens, and after scanning, we added scan parameters to the metadata (Supplementary Table 1). Using the AntCat[38] API in R 4.4.2, we checked the status of taxonomic descriptions for ant specimens included in Antscan. To add information relevant to taxonomy and systematics, we accessed a list of valid genera in AntWiki[56] and added this information to the metadata using R 4.4.2. We then preliminarily sorted specimens into scan trays by size, each tray containing 48 vials for automated scanning. To optimize the sorting step before scanning, specimens within one tray should have about the same size within the tube. That way, they can be assigned conveniently to the different available magnifications, and if necessary, the number of height steps can be estimated for the tray. We labeled vials with a code to trace each specimen back to its respective metadata. Tube labels were then the basis for file names during scanning. To make this physical identifier as legible and permanent as possible, flat snap-cap plastic tubes, ideally with a machine-written code attached to the top of the lid, have proven useful. All specimens are being stored indefinitely at OIST or have been returned to their owners or managing institutions.

### High-throughput synchrotron micro-CT

The specimens were scanned within two campaigns at the IPS Imaging Cluster at the KIT Light Source. Due to the available access, two different beamlines were used. During both campaigns, the same high-throughput tomography experimental station was employed, ensuring identical detector systems, available magnifications, scanning resolutions and sample exchange robotics (Advanced Design Consulting USA, Inc.). This guarantees the comparability of datasets at a given magnification setting within each measurement campaign.

In the first campaign, a set of samples was investigated using a parallel polychromatic X-ray beam produced by a 1.5-T bending magnet, spectrally filtered by 0.5 mm aluminum to remove low-energy components from the beam[57]. The resulting spectrum peaked at about 17 keV and a full width at half maximum bandwidth of about 10 keV. In the second campaign, the samples were scanned at the IMAGE beamline[58] by using a beam diffracted by a double multilayer monochromator. The measurements were performed at a magnetic field of the wiggler of 2.7 T, yielding a maximum flux density at 16 keV with an energy bandwidth of 2%. To reduce the heat load on the double multilayer monochromator, the beam was prefiltered with pyrolytic graphite.

Depending on their size, ants were scanned with magnifications of ×10, ×5 or ×2, resulting in an effective pixel size of 1.22 μm, 2.44 μm or 6.11 μm, respectively. An air-bearing rotary stage (RT150S, LAB Motion Systems) served for sample rotation. A fast indirect detector system consisting of a scintillator (×10: 13 μm LSO: Tb; ×5: 25 μm LSO: Tb; ×2: 200 μm LuAG: Ce), a double objective white beam microscope[59] (Optique Peter) and a 12-bit pco.dimax high-speed camera (Excelitas PCO GmbH) with 2,016 × 2,016 pixels was employed. For each scan, 200 dark-field images, 200 flat-field images and 3,000 equiangularly spaced radiographic projections in a range of 180° were taken with exposure times between 6.25 ms and 25 ms, resulting in scan durations between 21 s and 85 s. If the ants were too large for the field of view, additional height steps were scanned. In total, we acquired 3,998 scans (×10: 165; ×5: 3,026; ×2: 807) for 2,188 individual ants plus 31 outgroup Hymenoptera (×10: 149; ×5: 1,831; ×2: 239).

The control system 'concert' 0.31.0[60] served for automated data acquisition and online reconstruction of tomographic slices for data quality assurance. Online and final data processing included dark- and flat-field correction and, if applicable, phase retrieval of the projections based on the transport of intensity equation[61]. X-ray beam parameters for algorithms in the data processing pipeline were computed by syris[62], and the execution of the pipelines, including online tomographic reconstruction, was performed by the UFO 0.16 framework[63]. The final 3D tomographic reconstruction was performed by tofu 0.9[64] and additionally included ring removal, 8-bit conversion and blending of phase and absorption 3D reconstructions. Phase reconstruction was not performed for the previously iodine-stained specimens if their soft tissues still absorbed strongly. For the automatic processing of large sample series, our reconstruction software was extended by a sample and rotation axis finding procedure, which used a contrast measure to locate the top and bottom of a sample in two-dimensional (2D) projection. For these two positions, the program reconstructed several slices with different axis positions and computed a gradient-based measure to determine the correct axis. From having the correct axes for the top and bottom parts, the program could then detect whether the tomographic axis was perfectly aligned with the pixel columns and, if not, corrected for such bias.

Reconstructed tomogram image stacks are ordered from bottom to top. As conventions for how to read in data differ between software applications, the end user must confirm that the orientation of the scan is correct. Otherwise, the scan may appear mirrored, and it will be necessary to flip the scan to re-establish correct orientation.

### Laboratory micro-CT

The six largest specimens were scanned in the X-ray laboratory for CT and laminography of the IPS Imaging Cluster. For these, we employed a microfocus X-ray tube (XWT-225, X-RAY WorX, Garbsen), producing a conical, polychromatic X-ray beam from a solid tungsten target. We acquired the radiographic projections with a flatpanel detector (XRD 1621 CN14 ES PerkinElmer) featuring 2,048 × 2,048 pixels with a physical pixel size of 200 μm and a DRZ+ scintillator. Using the 'concert' control system, all components are positioned by a custom-built manipulator system. For the scans, the X-ray tube was operated with an acceleration voltage of 60 kV and a target power of 15 W. A total of 2,048 projections were acquired over an angular range of 360°. Each frame was exposed for 4 s. We placed the samples 69.7 mm from the source and the detector 1,630.5 mm downstream of the samples, resulting in a magnification of 24.4 and an effective pixel size of 8.2 μm. We performed two separate scans for each specimen to cover the whole animal. The 3D volumes were reconstructed with tofu[64].

### Post-processing of tomographic data

We employed an additional computer-vision pipeline in Python 3.10.12 using numpy 1.24.3 to, if necessary, merge height stages and generate 3D stacks of reduced file size cropped to the ant, 3D surface mesh models and sampled-down image stacks for 2D preview. We first stitched all individual CT scans for a single specimen to merge height steps based on Matte's Mutual Information metric in a multiresolution framework to sample exhaustively at lower resolutions before applying a gradient descent optimizer at finer scales implemented in Simple-ITK 2.3.1[65,66]. To stitch the height steps together, we first resampled the images to corresponding positions retrieved from registration and then, for merging, computed a weighted average gradient for the overlapping region where each of the two scans was given higher influence for the result the closer it was to the overlapping slice. For all datasets, we then used the deep learning feature in Biomedisa 23.09.1[21] to generate automated full-body segmentations. We trained this network with annotated data consisting of 12 randomly selected scans, which we manually presegmented, followed by Biomedisa's semi-automated segmentation[67]. We used the trained network to segment whole bodies

of all datasets and subsequently dilated these segmentations to mask a region around the ants using SciPy 1.11.4. We then used the minimum and maximum indices of the segmentations in all image planes to crop the volumetric images. With this workflow, we reduced the size of the online database by about 70%. Upon completion of merging and cropping, we retrained the neural network with 34 whole-body segmentations, of which 27 were used for training and 7 for validation. With this new deep neural network, we performed whole-body segmentations for all datasets. Using largest-islands selection to remove small parts, we isolated the ant to produce 3D surface meshes in STL format and rendered those in Paraview 2.12[68] for a 2D preview of the 3D habitus of the specimen. Finally, we sampled all stacks down to obtain a preview representation of the entire image stack.

### Further post-processing of exemplary datasets

For slices in Figs. 3 and 5, we opened tomograms in Fiji 2.15.1[69] or Amira 2020.2 to extract images with adjusted contrast to maximize visual tissue differentiation and separation from the background. For the surface renderings shown in Figs. 2–4, we employed Amira or 3D Slicer 5.8.1 for presegmentation of whole exoskeletons, individual sclerites and selected organs. Presegmented labels served as input for semi-automatic segmentation with Biomedisa[40,67]. When using Biomedisa, we imported the results back into Amira and corrected minor errors. For the 3D models shown in Fig. 5, we employed Amira's threshold tool to segment cuticle (threshold 105–255) and biomineral armor (200–255). We converted final label fields into polygon meshes, exported meshes as OBJ files and reassembled and smoothed them in CINEMA 4D R20. We also employed CINEMA 4D for artificial coloring, final image rendering and animation.

### Biomedisa instance of the Antscan database

The tomographic datasets of Antscan are stored and archived at KIT's Scientific Computing Center and linked with the Biomedisa platform. The metadata are stored in Biomedisa's MySQL database and backed up on an external storage system[70].

As datasets will be processed to generate new data and metadata of specimens that may change in the future (such as updated taxonomic status), we designed the database to enable such changes. All datasets can be freely accessed and downloaded. Selected registered Biomedisa users may act as administrators and can assign members of the database that have editing and annotation rights to enable modification of database entries. All registered Biomedisa users can perform semi-automated and automated segmentation analysis and may append their own data, such as segmentation results, to the specimens in their Biomedisa workspace.

### RADAR4KIT instance of the Antscan database

Sustainability of the entire Antscan database is ensured by providing a long-term-secured public location for all files including DOIs and metadata on the RADAR4KIT repository of Karlsruhe Institute of Technology (KIT). An overview can be directly accessed via https://radar.kit.edu/radar/en/search?query=antscan. We uploaded postprocessed tomograms from KIT's Scientific Computing Center to RADAR4KIT using a dedicated API and included metadata. We appended DOIs to the metadata in other locations to further enhance the accountability of each dataset generated within Antscan.

### Reporting summary

Further information on research design is available in the Nature Portfolio Reporting Summary linked to this article.

## Data availability

All processed Antscan tomograms are stored at the Large Scale Data Facility (LSDF) of KIT's Scientific Computing Center (SCC) and integrated into Biomedisa for public access under a CC BY 4.0 license (https://biomedisa.info/antscan/). Additionally, all datasets are archived under a CC BY 4.0 license at the RADAR4KIT research data repository (https://radar.kit.edu/radar/en/search?query=antscan). Both versions of the database and all metadata and DOIs linked to datasets are accessible via https://www.antscan.info. All metadata are also presented in the Supplementary Information. Raw 2D X-ray projections, which are not typically used directly for research, are further stored at KIT and will be made accessible upon reasonable request.

## Code availability

We provide code and instructions to postprocess micro-CT data on GitHub under a EUPL-1.2 license (https://github.com/julesforfools/Antscan). Within the same repository, we provide the code used to update metadata. Under a CC BY 4.0 license, we further provide the deep neural network used to segment, crop and visualize ant datasets in the RADAR4KIT repository (https://doi.org/10.35097/163wd8uzky1vh1bp) and on Biomedisa (see 'Antscan' on https://biomedisa.info/gallery/). This trained network is intended to be reused or retrained within the Biomedisa deep learning framework (https://github.com/biomedisa/biomedisa).

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

## Acknowledgements

Through T.B., research at KIT was supported by the projects HIGH-LIFE (grant no. 05K2019) and SMART-Morph (grant no. 05K2022) via the German Ministry for Research and Education (BMBF). We gratefully acknowledge the data storage service SDS@hd supported by the Ministry of Science, Research and the Arts Baden-Württemberg (MWK) and the German Research Foundation (DFG) through grant no. INST 35/1503-1 FUGG. J.K., F.H.G., F.A., A.R., A.C.-F., L.A., S.G., E.T., A.M. and E.P.E were supported by funding from the Okinawa Institute of Science and Technology Graduate University. The Japan Society for the Promotion of Science (JSPS) supported the work through KAKENHI grant nos. 18K14768 (F.H.G.), 21K06326 (F.H.G.), 24K01785 (E.P.E.) and 22KJ3077 (L.A.). E.P.E. and M.O. thank the Ministry of Environment of Japan for the Environment Research and Technology Development Fund (JPMEERF20234G01) of the Environmental Restoration and Conservation Agency. P.D.L. has received funding from the Australian Research Council via the ARC Training Centre for Multiscale 3D Imaging, Modelling, and Manufacturing (M3D Innovation, grant no. IC 180100008). S.C. was supported by HUN-REN Hungarian Research Network and by the National Research, Development, and Innovation Fund grant no. K 147781. R.M.F. was supported by the Conselho Nacional de Desenvolvimento Científico e Tecnológico (CNPq) (grant no. 301495/2019-0). L.S. was funded by the Deutsche Forschungsgemeinschaft (DFG, German Research Foundation), grant no. 502787686. P.G.H. was funded in part by the Critical Ecosystem Partnership Fund (CEPF). The Critical Ecosystem Partnership Fund is a joint initiative of l'Agence Française de Développement, Conservation International, the European Union, the Global Environment Facility, the Government of Japan and the World Bank. A fundamental goal is to ensure civil society is engaged in biodiversity conservation. B.L.F. acknowledges support through NSF grant no. DEB-1932467. D.A.G. has benefited from the equipment and framework of the COMP-HUB and COMP-R Initiatives, funded by the 'Departments of Excellence' program of the Italian Ministry for University and Research (MIUR, 2018-2022 and MUR, 2023-2027). B.G. acknowledged funding from the Environment and Conservation Fund (Hong Kong), grant no. Nb. ECF 137/2020. R.A.K. was supported by FCT (grant no. UID/00329/2025). T.A.L. acknowledges support through NSF grant no. NSF IOS-2128304. T.R.S. and J.S.C. were supported by U.S. National Science Foundation grant no. DEB 1927161. The funders had no role in study design, data collection and analysis, decision to publish or preparation of the manuscript. E.S. was supported by the 'Juan de la Cierva' program (JDC2024-054485-I) financed by the MICIU/AEI/10.13039/501100011033 and FSE+. We greatly appreciate P. S. Ward and the late C. Peeters for their guidance during the planning of the Antscan initiative and A. Cecilia for her great assistance at the IMAGE beamline and for contributing text to the Methods section. We would like to thank K. Soltau, S. Hofmann, S. Pollo Vázquez and the RADAR team at FIZ Karlsruhe, as well as J. Schwab, K. Simianer and the team at the KIT Library for their ongoing support and their contributions to the implementation of automated data processing within the RADAR4KIT Research Data Repository. We acknowledge J. Moren and the Scientific Computing and Data Analysis Section at OIST for providing resources and guidance to develop code. The original 3D models of the equipment shown in Fig. 1 were kindly provided by U. Herberger (KIT-IBPT; KIT Light Source), AVS|US Inc. (sample exchange robot), LAB Motion Systems (rotary stage), Optique Peter (detector system) and Excelitas PCO GmbH (high-speed camera). We acknowledge the KIT Light Source for provision of instruments at their beamlines, and we would like to thank the Institute for Beam Physics and Technology (IBPT) for the operation of the storage ring, the Karlsruhe Research Accelerator (KARA).

## Author contributions

J.K., F.H.G., T.F., P.D.L., J.J.B., R.M.F., L.S., G.Z., T.B., E.P.E. and T.v.d.K. conceived and conceptualized Antscan. F.H.G., A.C.-F., J.J.B., S.C., M.D., O.E., R.M.F., G.F., B.L.F., J.F.F., The GAGA Consortium, F.G., K.G., S.d.G., D.A.G., B.G., P.G.H., R.A.J., R.A.K., R.S.L., T.A.L., C.L., M.O., H.R., J.R., E.S., L.S., T.R.S., J.Z.S., J.S-C., C.T., L.T., S.Y., M.Y., G.Z., J.Z. and E.P.E. contributed material and contributed to the specimens included in Antscan. J.K., F.H.G., F.A., A.R., L.A., S.G., E.T. and T.v.d.K. prepared material for the micro-CT experiments. J.K., T.F., A.R., E.H., J.O., M.Z. and T.v.d.K. performed the micro-CT experiments. J.K., T.F., P.D.L., S.B., E.H., J.H., A.M., C.S., M.Z. and T.v.d.K. participated in reconstructing and/or processing the data. J.K., T.F., P.D.L. and C.S. built online database infrastructure. J.K., P.D.L., A.R., A.C.-F., L.A., S.G. and T.v.d.K. performed 3D reconstructions to include in the manuscript. E.P.E. and T.v.d.K. jointly supervised the work. J.K., E.P.E. and T.v.d.K. wrote the initial manuscript and created the figures. All authors extensively commented on and revised the manuscript at all stages.

## Competing interests

The authors declare the following competing interests: P.G.H. is the owner and CEO of the invertebrate consulting firm Afribugs CC. The other authors declare no competing interests.

## Additional information

**Correspondence and requests for materials** should be addressed to Julian Katzke, Francisco Hita Garcia, Evan P. Economo or Thomas van de Kamp.

# Reporting Summary

## Statistics

For all statistical analyses, confirm that the following items are present in the figure legend, table legend, main text, or Methods section.

| n/a | Confirmed | |
|---|---|---|
| ☒ | ☐ | The exact sample size (*n*) for each experimental group/condition, given as a discrete number and unit of measurement |
| ☒ | ☐ | A statement on whether measurements were taken from distinct samples or whether the same sample was measured repeatedly |
| ☒ | ☐ | The statistical test(s) used AND whether they are one- or two-sided<br>*Only common tests should be described solely by name; describe more complex techniques in the Methods section.* |
| ☒ | ☐ | A description of all covariates tested |
| ☒ | ☐ | A description of any assumptions or corrections, such as tests of normality and adjustment for multiple comparisons |
| ☒ | ☐ | A full description of the statistical parameters including central tendency (e.g. means) or other basic estimates (e.g. regression coefficient) AND variation (e.g. standard deviation) or associated estimates of uncertainty (e.g. confidence intervals) |
| ☒ | ☐ | For null hypothesis testing, the test statistic (e.g. $F$, $t$, $r$) with confidence intervals, effect sizes, degrees of freedom and $P$ value noted<br>*Give P values as exact values whenever suitable.* |
| ☒ | ☐ | For Bayesian analysis, information on the choice of priors and Markov chain Monte Carlo settings |
| ☒ | ☐ | For hierarchical and complex designs, identification of the appropriate level for tests and full reporting of outcomes |
| ☒ | ☐ | Estimates of effect sizes (e.g. Cohen's *d*, Pearson's *r*), indicating how they were calculated |

*Our web collection on statistics for biologists contains articles on many of the points above.*

## Software and code

Policy information about availability of computer code

| Data collection | We employed the control system concert 0.31.0 and the UFO 0.16 framework for data acquisition and online reconstruction of tomographic slices. Final reconstruction of tomographic data was done with tofu 0.9. All software packages are referenced in the Methods section. |
|---|---|
| Data analysis | Data were analyzed using R 4.4.2, Python 3.10.12, numpy 1.24.3 , Simple-ITK 2.3.1, SciPy 1.11.4, Biomedisa 23.09.1, Paraview 2.12, Fiji 2.15.1, Amira 2020.2, 3D Slicer 5.8.1 & CINEMA 4D R20. Code available at https://github.com/julesforfools/Antscan. |

For manuscripts utilizing custom algorithms or software that are central to the research but not yet described in published literature, software must be made available to editors and reviewers. We strongly encourage code deposition in a community repository (e.g. GitHub). See the Nature Portfolio guidelines for submitting code & software for further information.

## Data

Policy information about availability of data

All manuscripts must include a data availability statement. This statement should provide the following information, where applicable:

- Accession codes, unique identifiers, or web links for publicly available datasets
- A description of any restrictions on data availability
- For clinical datasets or third party data, please ensure that the statement adheres to our policy

All processed Antscan tomograms are stored at the Large Scale Data Facility (LSDF) of KIT's Scientific Computing Center (SCC) and integrated into Biomedisa for public access (https://biomedisa.info/antscan/). Additionally, all datasets are archived at the RADAR4KIT repository (https://radar.kit.edu/radar/en/search?

## Human research participants

Policy information about studies involving human research participants and Sex and Gender in Research.

| | |
|---|---|
| Reporting on sex and gender | n/a |
| Population characteristics | n/a |
| Recruitment | n/a |
| Ethics oversight | n/a |

Note that full information on the approval of the study protocol must also be provided in the manuscript.

# Field-specific reporting

Please select the one below that is the best fit for your research. If you are not sure, read the appropriate sections before making your selection.

☒ Life sciences          ☐ Behavioural & social sciences          ☐ Ecological, evolutionary & environmental sciences

For a reference copy of the document with all sections, see nature.com/documents/nr-reporting-summary-flat.pdf

# Life sciences study design

All studies must disclose on these points even when the disclosure is negative.

| | |
|---|---|
| Sample size | Our study does not deal with experimental series of similar samples, but presents a collection of diverse specimens. Therefore no sample size calculation was required. |
| Data exclusions | No data were excluded from the analyses. |
| Replication | The reproducibility of the experimental results is guaranteed by the description of the methods in the manuscript and by the data and code provided |
| Randomization | The study does not include experimental series but is based on individual specimens. |
| Blinding | This paper does not deal with experimental series of similar samples, which would require statistical analysis. Blinding was therefore not applicable here. |

# Reporting for specific materials, systems and methods

We require information from authors about some types of materials, experimental systems and methods used in many studies. Here, indicate whether each material, system or method listed is relevant to your study. If you are not sure if a list item applies to your research, read the appropriate section before selecting a response.

## Materials & experimental systems

| n/a | Involved in the study |
|---|---|
| ☒ ☐ | Antibodies |
| ☒ ☐ | Eukaryotic cell lines |
| ☒ ☐ | Palaeontology and archaeology |
| ☐ ☒ | Animals and other organisms |
| ☒ ☐ | Clinical data |
| ☒ ☐ | Dual use research of concern |

## Methods

| n/a | Involved in the study |
|---|---|
| ☒ ☐ | ChIP-seq |
| ☒ ☐ | Flow cytometry |
| ☒ ☐ | MRI-based neuroimaging |

# Animals and other research organisms

Policy information about studies involving animals; ARRIVE guidelines recommended for reporting animal research, and Sex and Gender in Research

| | |
|---|---|
| Laboratory animals | The study does not involve laboratory animals |
| Wild animals | The study does not involve wild animals. |
| Reporting on sex | The findings themselves are not sex-specific. However, while we have included male ants, a natural female bias arises because all ant workers are female. The sex of each specimen is specified in the metadata accompanying each tomogram and listed in the Supplementary Data. |
| Field-collected samples | No new samples were collected for this study. All specimens originate from standing insect collections as detailed in the metadata accompanying each tomogram and listed in the Supplementary Data. |
| Ethics oversight | All study specimens are ethanol-fixed insects from standing insect collections, which do not require ethical approval or guidance. All export and import of specimens was conducted according to national and international guidelines through permits and institutional collaborations. |

Note that full information on the approval of the study protocol must also be provided in the manuscript.

