## [Peer Review File · Nature Methods]

High-throughput phenomics of global ant diversity

Corresponding Author: Dr Julian Katzke

Version 0:

Decision Letter:

Our ref: NMETH-RS58707

9th Apr 2025

Dear Dr. Katzke,

Thank you for submitting your revised manuscript "High-throughput phenomics of global ant diversity" (NMETH-RS58707). Thank you also for your patience during the review. The manuscript has now been seen by the original referees and their comments are below. The reviewers find that the paper has improved in revision, and therefore we'll be happy in principle to publish it in Nature Methods, pending minor revisions to satisfy the referees' final requests and to comply with our editorial and formatting guidelines.

Please also note that I had hoped to receive input from a reviewer with expertise in microCT, which unfortunately hasn't happened. I will however have someone quickly check this once the manuscript is revised.

I would also like to ask you to make it very clear from the beginning that the purpose of the manuscript is to present a resource. I also suggest de-emphasizing the microCT pipeline a bit.

Reviewer #1 provided thorough input on the manuscript file. However, the file is too large to send as an attachment via our manuscript tracking system. I'll share it with you in a separate message. Please let me know if you have trouble accessing it.

TRANSPARENT PEER REVIEW

ORCID

Best regards,
Nina

Nina Vogt, PhD
Senior Editor
Nature Methods

Reviewer #1 (Remarks to the Author):

The manuscript by Katzke et al. provides an overview of a massive, novel phenomic dataset of ants, generated with high-throughput micro-computed tomography (microCT). This dataset spans the phylogenetic and morphological diversity of ants, with 3D whole-body scans of 800 species representing about 60% of all extant ant genera. This is a truly impressive dataset that, if truly accessible, would lead to significant advances in the field of myrmecology.

A disclaimer: I am not an expert on micro-computed tomography and I recommend that this manuscript is reviewed by at least one other person with intimate familiarity with this technology.

I see three areas for improvement of this manuscript.

First, the writing is somewhat unclear about what is the "method" being presented. It appears to me that the main subject here is the dataset of 3D scans. However, in several places in the text, including the introduction and large sections of the discussion, the reader's attention is directed towards the high-throughput microCT workflow. This makes the reader think the goal of the paper is to present this workflow and somehow make it more accessible to other researchers. That does not appear to be the case, and so focus on the workflow, however novel it may be, is potentially confusing for the reader; it was for me. My comments directly on the manuscript text highlight some of these areas. The authors should mention the workflow used to generate the scans, as it emphasizes the value of the dataset, but the focus should be on the dataset itself and how it can be used. The workflow can be a methods note in itself, but here it is not being democratized in any meaningful way, and therefore not serving the purpose of "providing a tool" per the aims of the journal. Refocusing the narrative on the scans is the main area for improvement I could discern.

Second, accessibility and copyright. "Free" access is mentioned in the abstract and some practical aspects of data access are explained in the "Biomedisa instance of the Antscan database" section. Because the main resource reported here is the wealth of anatomical scan data, I believe that more focus in the main manuscript text should be given to this and additional details are needed. It would be good to have a primer on how it can be used, what the attribution expectations are for users, formal copyright statement, and other rules around proper and improper use explicitly stated. Ideally, anyone anywhere would be able to use the scans for their research citing this paper, similarly to how published sequence data is treated. This should be made more explicit in the main body of the manuscript.

Further, the success and ultimate value of this resource to myrmecology will depend on the Antscan user base. An explanation of the lingo connected to micro computed-tomography and more resources connected to the use cases presented would help develop this base. As someone with superficial familiarity with this technology, I would have found an introduction to the computer tomography technology and data processing concepts helpful. Perhaps a box of definitions? This should be aimed at both someone trying to understand the methods used to generate the scans, as well as a researcher who is wishing to use them in their morphological and/or comparative work. Furthermore, providing the reader with or pointing to resources on how to accomplish the use cases presented would be a powerful demonstration of how this type of dataset can advance the field. The workflow for a detailed segmentation such as the Eciton worker or screening for biomineralization would enable the reader to see how they can use Antscan as a resource.

Sincerely,
Marek Borowiec

Reviewer #1 (Remarks on code availability):

I was not able to access the code under the link provided above due to access expired error.

Reviewer #2 (Remarks to the Author):

The authors have developed a robust pipeline for high throughout computed tomography of ants. They find that they can in short time frame recover phylogenetically informative morphological features from specimens placed in an array and scanned. This is incredibly innovative and will be transformative for the field of morphology. I think that the paper could please provide information about the two phylogeny figures, which trees these were and how they colored the branches to indicate sampling

efforts. This should be mentioned in the methods and or results. The references cited are appropriate and I honestly am blown away but this remarkable paper! I would say this paper should be accepted with very minor revisions.

Version 1:

Decision Letter:

9th Jan 2026

Dear Dr Katzke,

Thank you for your patience. I am pleased to inform you that your Resource, "High-throughput phenomics of global ant diversity", has now been accepted for publication in Nature Methods. The received and accepted dates will be November 20th, 2024 and January 9th, 2026. This note is intended to let you know what to expect from us over the next month or so, and to let you know where to address any further questions.

Over the next few weeks, your paper will be copyedited to ensure that it conforms to Nature Methods style. Once your paper is typeset, you will receive an email with a link to choose the appropriate publishing options for your paper and our Author Services team will be in touch regarding any additional information that may be required. It is extremely important that you let us know now whether you will be difficult to contact over the next month. If this is the case, we ask that you send us the contact information (email, phone and fax) of someone who will be able to check the proofs and deal with any last-minute problems.

Authors may need to take specific actions to achieve compliance with funder and institutional open access mandates.

If your research is supported by a funder that requires immediate open access (e.g. according to [Plan S principles](https://www.springernature.com/gp/open-science/plan-s-compliance) or the [NIH public access policy](https://www.springernature.com/gp/open-science/us-federal-agency-compliance)) then you should select the gold OA route, and we will direct you to the compliant route where possible. Because authors warrant under our subscription licensing terms that they haven't committed to licensing any version of their article under a licence inconsistent with the terms of our agreement – including the applicable embargo period – publication under the subscription model isn't suitable for authors whose funders require no embargo.

If you are active on Twitter/X or Bluesky, please e-mail me your and your coauthors' handles so that we may tag you when the paper is published.

Best regards,
Nina

Nina Vogt, PhD
Senior Editor
Nature Methods

** Visit the Springer Nature Editorial and Publishing website at http://editorial-jobs.springernature.com?utm_source=ejp_NMeth_email&utm_medium=ejp_NMeth_email&utm_campaign=ejp_Nmeth for more information about our career opportunities. If you have any questions please click [here](mailto:editorial.publishing.jobs@springernature.com).

Dear Nina,

We are pleased to submit a revised version of our article “High throughput phenomics of global ant diversity” (NMETH-RS58707) to *Nature Methods*. We have carefully edited the manuscript to address the reviewers’ comments. We have followed advice to edit and more clearly highlight the data resource itself as the main contribution of the paper. Our article is a “resource” paper. However, we do still discuss the methods, as they are key to understanding the resource, and we feel the study design itself is transformative and may inspire future projects.

Please find below a point-by-point response to the comments of both reviewers, including the comments that Reviewer #1 embedded in the PDF. In addition, we made a few minor corrections of our own. To facilitate following all revisions, we provide line numbers that match the revised manuscript with tracked changes we are submitting.

As requested, we have lifted the restricted access to antscan.info, so all Antscan tomograms are now publicly available. The automated publication process via RADAR will be completed in about two days. All links and DOIs listed in the manuscript and supplementary material are final.

Thank you again for your work on this manuscript.

Best wishes

Julian, Francisco, Evan & Thomas

Point-to-point response

Reviewer #1:

“The manuscript by Katzke et al. provides an overview of a massive, novel phenomic dataset of ants, generated with high-throughput micro-computed tomography (microCT). This dataset spans the phylogenetic and morphological diversity of ants, with 3D whole-body scans of 800 species representing about 60% of all extant ant genera. This is a truly impressive dataset that, if truly accessible, would lead to significant advances in the field of myrmecology.”

RESPONSE: Thank you for your positive comments and thoughtful review. We have addressed your comments below and via changes to the manuscript.

„First, the writing is somewhat unclear about what is the "method" being presented. It appears to me that the main subject here is the dataset of 3D scans. However, in several

places in the text, including the introduction and large sections of the discussion, the reader's attention is directed towards the high-throughput microCT workflow. This makes the reader think the goal of the paper is to present this workflow and somehow make it more accessible to other researchers. That does not appear to be the case, and so focus on the workflow, however novel it may be, is potentially confusing for the reader; it was for me. My comments directly on the manuscript text highlight some of these areas. The authors should mention the workflow used to generate the scans, as it emphasizes the value of the dataset, but the focus should be on the dataset itself and how it can be used. The workflow can be a methods note in itself, but here it is not being democratized in any meaningful way, and therefore not serving the purpose of "providing a tool" per the aims of the journal. Refocusing the narrative on the scans is the main area for improvement I could discern."

RESPONSE: We agree the aspect being "democratized" here is the data resource which now will allow labs around the world to study ant anatomy in 3D with only access to a computer. This is indeed why we submitted the paper under the "Resource" category. Accordingly, we have shifted the emphasis towards Antscan as a resource in several key sections of the manuscript (lines 139-142, 164-165, 349-355, 380-381, 395-410). However, we feel the design of the project that enabled the acquisition of these data is novel and interesting beyond the value of these data on ants because it can be applied across a wide variety of taxa. We believe that this paper could stimulate the collaboration of biodiversity collections with imaging facilities and adopt automation techniques to digitize the anatomical diversity of life on earth. We discuss the current impediments and technological limitations and lay out the necessary steps to overcome these for an implementation of our chosen workflow beyond ants. Thus, because the workflow enabled the resource, we feel strongly that the workflow should be discussed and its future potential highlighted.

"Second, accessibility and copyright. "Free" access is mentioned in the abstract and some practical aspects of data access are explained in the "Biomedisa instance of the Antscan database" section. Because the main resource reported here is the wealth of anatomical scan data, I believe that more focus in the main manuscript text should be given to this and additional details are needed. It would be good to have a primer on how it can be used, what the attribution expectations are for users, formal copyright statement, and other rules around proper and improper use explicitly stated. Ideally, anyone anywhere would be able to use the scans for their research citing this paper, similarly to how published sequence data is treated. This should be made more explicit in the main body of the manuscript."

RESPONSE: The data is open and shared via the Biomedisa and RADAR repositories under a CC BY 4.0 license. This means that the scans can be used for research or any other purpose with only the condition that the source (i.e., this paper) should be cited. The RADAR repository will ensure long-term accessibility similar to NCBI for sequence

data. We followed the advice, particularly by rewriting the results section “The Public Antscan Repositories” (lines 278-310) to include proper attribution and licensing information. We also updated the Data Availability Statement.

“Further, the success and ultimate value of this resource to myrmecology will depend on the Antscan user base. An explanation of the lingo connected to micro computed-tomography and more resources connected to the use cases presented would help develop this base. As someone with superficial familiarity with this technology, I would have found an introduction to the computer tomography technology and data processing concepts helpful. Perhaps a box of definitions? This should be aimed at both someone trying to understand the methods used to generate the scans, as well as a researcher who is wishing to use them in their morphological and/or comparative work. Furthermore, providing the reader with or pointing to resources on how to accomplish the use cases presented would be a powerful demonstration of how this type of dataset can advance the field. The workflow for a detailed segmentation such as the Eciton worker or screening for biomineralization would enable the reader to see how they can use Antscan as a resource.”

RESPONSE: Although we understand the reviewer’s perspective, we feel that a tutorial on workflows for processing scan data may not be appropriate here. This is because our processing methods beyond the computer vision pipeline are not novel and are general to any CT scan data, and are routinely used, and thus, we assume it would not be appropriate for Nature Methods. Moreover, there are such tutorials available elsewhere, for example associated with open software such as 3D Slicer (slicer.org) and many others. We describe the methods used for the highlighted examples in this paper (lines 538-548) and have revised the respective paragraphs to make them more accessible to readers less experienced with the methods (lines 130-137, 231-238).

We have emphasized that the citations in the discussion regarding CT processing are also educational (lines 398-399) and we have added tutorials to the Antscan homepage (<https://www.antscan.info/tutorials>) that outline how to recreate the segmentations used in Figures 3 and 5 exclusively with the open-source software 3D Slicer.

“I was not able to access the code under the link provided above due to access expired error.”

RESPONSE: We apologize for these shortcomings and have since then created a GitHub Repository with code used as well as instructions how to run it. We edited the Code availability statement accordingly (lines 604-606).

Reviewer #1 (comments in the manuscript file)

Line 31: “It would be good to include the number of genera represented here also.”

RESPONSE: Added number of genera to the abstract (81).

Lines 80-82: “Why this matters”

RESPONSE: We added a statement to emphasize that Synchrotron micro-CT can be a key technology for the pursuit of digitizing phenomic diversity addressing the challenges outlined in the two paragraphs above (130-133):

“Synchrotron-based micro-CT offers the potential to perform standardized 3D imaging of large numbers of ethanol-preserved collection samples without additional invasive preparation.”

Lines 87-89: “This signals the main focus of the paper is the workflow to rapidly generate 3D morphological data but in fact it is the generated data itself? You do not actually give readers the tools to generate synchrotron scans more easily themselves.”

RESPONSE: We addressed and rewrote this in the context of the main point raised by the reviewer (139-140). The revised paragraph now explicitly identifies the Antscan database as the key outcome of the initiative:

“Within the ‘Antscan’ initiative, we created a massive open resource for research on the phenotypic diversity of ants”

Lines 87: “Is Antscan the workflow for imaging, or data generated, or both? What sort of data is generated? Scans, 3D models, or segmentations? Or all of them? What is the difference between these different levels of data processing, and how can each be used?”

RESPONSE: We now clarify that the Antscan initiative comprises both the process (method) and the resulting large-scale resource (164-165).

Line 135: “Exactly 342 genera as of December 30, 2024. Cite AntCat.org.”

RESPONSE: Revised to currently 343 including updated accession date (185).

Line 151: “How does the number 186 (“species w/ genome”) in Figure 2 relate to the numbers here?”

RESPONSE: The section has been rewritten to lead with the number of species for improved clarity (200-202).

Lines 181-187: “This paragraph is very jargony. Some background on what are iodine-treated specimens and why they are created would be helpful to the reader. Similarly for phase retrieval.”

RESPONSE: Rewritten to make the paragraph understandable for a broader audience (lines 231-238).

We express this with three phrases that are hopefully easy to digest:

“We included 132 ants that were originally prepared for a laboratory micro-CT project. These specimens were stained with iodine to enhance soft tissue contrast. Due to their high X-ray absorption, phase retrieval was not applicable in these cases.”

Lines 213-215: “Consider breaking this sentence into two.”

RESPONSE: Done (263-265).

Lines 259-261: “How? Show me, don't tell me.”

RESPONSE: This sentence did not belong into this section and was copy/pasted from the discussion by error. We removed it.

Lines 280-281: “surface of the cuticle”

RESPONSE: Corrected (332).

Line 298: “This opening of the discussion section makes it unclear if the method presented is the digitization workflow or the scan database.”

RESPONSE: We revised the paragraphs to more clearly highlight the resource. While retaining the broader context of integrating biodiversity collections with high-throughput micro-CT, we now focus the discussion of the pipeline specifically on how it contributes to the generation of the Antscan resource (349-381).

For example:

“With Antscan, we provide an open resource that spans the morphological and anatomical diversity of ants.”

“We explicitly address points of criticism in the accessibility of and credit for digitized biodiversity; our database is open, and the metadata identifies all contributors.”

“Together, the pipeline and the resulting database provide a foundational resource for advancing comparative morphology in the digital era.”

Lines 302-303: “But this is not the “method” you are presenting in this publication.”

RESPONSE: We revised the topic sentence of this paragraph to tailor more towards the connection between high-throughput synchrotron microscopy and the Antscan resource (367):

“High-throughput synchrotron micro-CT is a key technology to effectively overcome the bottleneck of acquisition times in micro-CT.”

Line 312: “What is the focus of this paper? The scanning pipeline, or data generated? Or both? I think that the challenge of generating the data does underscore the value of the scans , but this point can be made more succinctly.”

RESPONSE: According to the reviewer suggestions, we have revised this paragraph to emphasize that the scale of the Antscan database depended on these efforts beyond

micro-CT scanning (367-381). We intend that this and the previous paragraphs together emphasize the necessary building blocks to create such a resource, the resource being the focus of the paper.

Lines 360-362: “I think a brief discussion of use cases and this latent potential of what Antscan could enable in the future, would both be very interesting.”

RESPONSE: The preceding paragraph has been rewritten to provide more detail on the potential use cases of the Antscan datasets (395-411).

Reviewer #2:

“The authors have developed a robust pipeline for high throughout computed tomography of ants. They find that they can in short time frame recover phylogenetically informative morphological features from specimens placed in an array and scanned. This is incredibly innovative and will be transformative for the field of morphology.”

RESPONSE: Thank you for your positive comments.

„I think that the paper could please provide information about the two phylogeny figures, which trees these were and how they colored the branches to indicate sampling efforts. This should be mentioned in the methods and or results.”

RESPONSE: We now reference the sources of the phylogenies in the according figure captions (212, 346) and would otherwise leave the main text unaltered for the sake of brevity.

From the editorial team:

“Please consider making “Raw X-ray projections and other image data” available in a publicly accessible repository, or explain to the editor why the data can only be made available from the authors on request.”

RESPONSE: We apologize if this was not clear. The main image product being shared in this study is the 8-bit tomogram that blends phase contrast and absorption contrast into a single image, these are available on public repositories. The raw 2D X-ray projections are used only as input for tomographic reconstructions, these are not normally shared in datasets or papers that use CT data and they are (to our knowledge) not useful for research purposes until they are turned into volumetric tomograms. Raw projections are not even typically saved by researchers who use lab scanners as the tomographic reconstruction is handled within the machine’s software. Due to their size, posting the projections would increase the size of the shared data by ~200%, which would be a challenge for the repositories we are using.

We have rephrased this section to make this more clear:

“

All processed Antscan tomograms are stored at the Large Scale Data Facility (LSDF) of KIT's Scientific Computing Center (SCC) and integrated into Biomedisa for public access under a CC BY 4.0 license (<https://biomedisa.info/antscan/>). Additionally, all datasets are archived under a CC BY 4.0 license at the RADAR4KIT research data repository (<https://radar.kit.edu/radar/en/search?query=antscan>). Both versions of the database and all metadata and DOIs linked to datasets are accessible via <https://antscan.info>. All metadata is also presented in the supplementary information to this publication. *Raw 2D X-ray projections, which are not typically used directly for research, are further stored at KIT and will be made accessible upon reasonable request.*

“

Additional minor corrections to the manuscript

- Changed several uses of 'supplemental' to 'supplementary' to conform with the journal
- Changed 'manifests' to 'is manifested' (88)
- Rewrote several uses of 'significant' in a non-statistical context
- Changed grammar to avoid using 'Especially for...' (115)
- Rewrote description of ant lifestyle to avoid stating that males are an ant caste
- Edited unpublished work citations (159, 202, 409)
- Rewrote elements of the Figure 3 caption to enhance readability
- Added 'education, and art' to emphasize the broad potential of the Antscan resource (422)
- Added version numbers for all software
- Corrected misspelling of 3D Slicer (541)
- Edited user rights in Biomedisa for clarity (556-560)
- Removed all mentions of access restrictions or preliminary access
- Added funders non-involvement statement
- Added acknowledgement to the reviewers
- Added acknowledgement to RADAR team
- Added acknowledgement to Scientific Computing at OIST
- Separated References and Methods-only References
- Added reference 57 to cite a new publication on the IMAGE beamline
- Corrected Biomedisa GitHub link
- Added Irene Villalta to the GAGA Consortium Authors
- Added "the late" to the acknowledgement of Christian Peeters
- Updated OIST funding acknowledgements

- Explicitly outlined in Methods how taxonomic and systematic information was updated and added using AntCat and AntWiki (436-440). Code added to Antscan GitHub
- Added “Legend” sheet to explain columns in Supplementary Table S2
- Slightly edited Fig. 1 for aesthetic appeal
- Corrected text alignment in Fig. 5
- Added GAGA Consortium at the end of the manuscript following guidelines

**High-throughput phenomics of global ant biodiversity**

Keywords: 3D data, digitization, tomography, soft tissue preservation, insects, Formicidae,
anatomy, morphology, database

6 **Authors:**

[revised manuscript text omitted]

ant sizes in the collection (Fig. 3, Material and Methods). Specimens exceeding the field of view
vertically were scanned in several height steps.

In addition to absorption contrast, synchrotron measurements also exploit phase contrast due to
the partial transverse coherence of the radiation. Phase contrast arises during the free-space
propagation of the transmitted wave field, which already after short propagation distances
enhances the tissue boundaries in the measured images (so-called edge enhancement³⁹). This is
particularly important for soft tissues such as muscles, which absorb almost no radiation. A
phase retrieval algorithm then converts the edge enhancement into distinguishable tissue
contrast. To obtain good overall contrast for both higher absorbing parts like the exoskeleton and
soft tissues, the primary datasets of the Antscan database are blended volumes, composed of one
volume obtained by standard 3D reconstruction of the measured images and the other by
additional phase retrieval applied prior to the 3D reconstruction.

As an exception to the blended-volume datasets that comprise most of the Antscan scans, we
deviated for 132 previously iodine-treated ants and the six largest ants we gathered. Due to the
strong absorption of their soft tissues, most of the iodine-stained specimens were reconstructed
without phase-retrieval. For the six largest specimens, which exceeded even the largest field of
view available at the synchrotron setup in diameter, we performed iodine-staining and scanned
them with a laboratory micro-tomograph. Although differing in imaging, we could incorporate
these specimens in all downstream processing steps.

While the original tomograms are saved as 32-bit data, we converted and further processed the
tomograms as 8-bit TIFF stacks. When required, we used a computer-vision workflow to merge
individual height steps automatically into a combined volumetric dataset, showing the entire ant
body. Employing a neural network using Biomedisa^{21,40}, we further crudely segmented datasets
automatically and cropped all tomograms by removing background and thus reducing file size
considerably. This automated processing of all data underlines the suitability of high-throughput
synchrotron micro-CT for future large-scale analyses using computer vision and machine
learning.

Since the data acquisition and processing parameters were identical for the sample subsets
scanned at a given magnification at a beamline (see Methods), the reconstructed gray values of
the corresponding tissues within these subsets are equivalent (except for few iodine-treated

samples), ensuring comparability and facilitating the application of machine learning and
computer vision approaches.

**Fig. 3: Exemplar images representing different magnifications for different-sized ant workers. Top:** 3D
models of different ant workers scanned within Antscan depicted to illustrate operational scales. When scaled to an
A4 page, the ants appear in their original size. **Bottom:** Image slices from the heads of four ant species acquired
with different magnifications. From left to right: *Paraponera clavata* stained with iodine and scanned using
laboratory micro-CT as one of the largest ants that did not fit inside the largest field of view of the synchrotron
setup; *Eciton hamatum* subsoldier as a large ant; *Gnamptogenys* aff. *continua* as a medium-sized ant; *Discothyrea*
*sexarticulata* as one of the smallest ants.

For quality assurance, we visually inspected all scans to check for specimen preservation quality.
Most ants were well preserved, but unavoidable when drawing from standing collections, but
some showed severe damage from soft tissue shrinkage and decay, probably caused by DNA
extraction, exposure to air during transport, handling, or storage. However, as the rigid
exoskeleton was generally not affected, many morphological or morphometric analyses can still
be applied. Overall, given the preservation quality, we conclude that both short- and long-term
stored ethanol-preserved material is generally suitable for non-destructive high-throughput
synchrotron micro-CT to digitally preserve 3D anatomy of invertebrates indefinitely.

Since samples were prepared and positioned manually beforehand and scanning proceeded
automatically, the partial truncation of some datasets could not be avoided. Legs and antennae of
larger specimens within a magnification category were particularly prone to be outside the field
of view during scanning. Future developments towards improved robotic setups and imaging

pipelines that can autonomously recognize the individual position of specimens during
measurements and readjust scanning positions accordingly will likely resolve such minor issues.

The Public Antscan Repositories

All tomograms acquired within Antscan and their associated metadata are provided publicly in
open repositories that can be accessed via the Antscan website (<https://antscan.info> *Password
for review status: antscanreview*) or directly (<https://biomedisa.info/antscan>,
<https://radar.kit.edu/radar/en/search?query=antscan> *Data on the system is not finalized but can
be directly accessed from individual links provided in Supplemental Table S2*). The interface
for the primary, interactive repository was created on the image analysis and segmentation
platform Biomedisa⁴⁰. A search function offers the possibility to explore the database for specific
keywords in the metadata. Individual datasets are represented on the top page by 3D surface
previews based on neural network automatic segmentation. The ant datasets then feature more
previews for image slices and an overview of the insect habitus as an interactive 3D model of the
surface mesh. Apart from the tomographic data, each ant is accompanied by comprehensive
metadata, including taxonomic ranks, ecological parameters, and unique specimen identifiers
(Supplemental Table S2). We also provide information on specimen locality, further
geographical details, habitat, and whether a sequenced genome is associated with the specimen.
Metadata further provides the information required to identify and acknowledge curators, other
contributors, and host institutions responsible for the individual specimens.

The public Antscan database infrastructure is curated, aims to provide an interactive experience,
and is sustainable. Integration into the Biomedisa interface offers the possibility for registered
users to apply methods for semi-automatic and automatic volumetric image segmentation with
the Biomedisa app, which can be directly applied to the datasets. Datasets that are being
processed can be easily shared with other Biomedisa users. In addition to Biomedisa, we ensure
the sustainability of the entire Antscan database by providing another long-term-secured location
for all files including metadata on the RADAR4KIT repository of KIT. The chosen format of a
mirrored, open database allows for smooth phylogeny/taxonomy-based online navigation and
assures public long-term availability of the Antscan data.

Exemplary use cases

To demonstrate the wealth of information contained in an individual ant scan, we segmented
exoskeletal elements, muscles, nervous tissues, sting apparatus, and digestive tract of a sub-
soldier of the South American army ant *Eciton hamatum* (Fig. 4). Such segmented anatomical
features can, e.g., be extracted and quantified for analyses, visualized as renderings (Fig. 2, 3 &
4a), animated (see Extended Data Video E1), and 3D-printed. Detailed digital examinations and
the extraction of morphological characters of single specimens already do and will likely lead to
new discoveries.

**Fig. 4: Renderings of an exemplary Antscan specimen (*Eciton hamatum* sub-soldier CASENT0744582)**
 **showing the segmented cuticle and tissues representative of the level of detail captured with synchrotron**
 **micro-CT. a: Full habitus of the ant with an animated, more life-like pose and colors inspired by photographs. b:**
 **Cuticle cut at the sagittal section revealing internal tissues with muscles in red occupying most of the internal space**
 **in an ant's body. c: Removing the muscles reveals the digestive tract (green) and the nervous system (blue). d-f:**
 **Zoomed-in renderings focusing on the ant brain, gut, and sting apparatus, respectively.**

Moreover, the vast amount of available specimen data facilitates large-scale comparative studies,
 e.g., to identify similarities and differences between lineages and to trace the evolution of traits

throughout the ant tree of life. In this context, not all scientific cases require extensive
segmentations of individual ants. Simple screening of the datasets may already quickly clarify
whether morphological features are present or absent in ant species or entire lineages. Here, this
is shown on the example of biomineral armor, an unusual insect trait first described for
*Acromyrmex echinator*⁴¹ and previously unknown in other ants. The Antscan data immediately
revealed that this trait is more widely distributed. A conspicuous highly absorbing layer on top of
the cuticle confirms that biomineral armor is in fact common among fungus-growing ants
(Myrmicinae: Attini) and scattered across the different agricultural systems that evolved in these
clades^{42,43} (Fig. 5). Within the attine ants, it appears to be absent in the genera *Atta*,
*Kalathomyrmex*, and *Mycocepurus*, which is consistent with an inferred secondary loss of
biomineral armor⁴¹. We have not found evidence of this trait in any species outside the Attini.
This example illustrates the potential for standardized high-throughput imaging datasets to
enable testing evolutionary and ecological hypotheses at larger scales without the acquisition of
new data.

**Fig. 5: Trait recognition with Antscan on the example of biomineral armor.** Comparative screening revealed
 that biomineral armor is very common among fungus-growing ants and was found in several attine species. The
 slices through the tomograms (right) show the armor as a thin but distinct coat of highly absorbing (brighter)
 material on top of the cuticle. In the simple thresholded 3D renderings (left) it is visualized as a beige color.

Discussion

With Antscan, we present a scalable design to meet the demands for large-scale 3D digitization
 of phenotypes from biodiversity collections. It hinges on first centralizing specimens to fit them
 to the high-throughput imaging setup, then scanning in a condensed timeframe, reconstructing

and processing CT-scans to finally provide them in an open-access format. We explicitly address
points of criticism in the accessibility of and credit for digitized biodiversity; our database is
open, and the metadata identifies all contributors^{8,11,44}.

Inherently, high-throughput synchrotron micro-CT effectively overcomes the bottleneck of
acquisition times in micro-CT. It is fast and provides well-resolved morphological datasets with
high contrast for both hard and soft tissues if specimens have been properly preserved. Including
file transfer, which currently constitutes a technological bottleneck in our scanning setup,
Antscan moved at a pace of ~25 scans per hour. Similarly resolved scans of ants using
conventional micro-CT would take about 12 hours each on average^{45,46}, with the additional delay
of individual scan setups. Extrapolating this to the 4,010 individual scans in Antscan, it would
have taken one laboratory micro-CT scanner operated around the clock more than six years to
obtain a similar dataset.

The Antscan workflow implementation does demand considerable effort and coordination. A
collaborative approach among global collections stakeholders is necessary, including meticulous
management of institutional and governmental requirements, such as in specimen transfer. This
engagement, however, translates into scientific value through accurate documentation of
specimens and accountability for contributors. The laborious process of sorting and preparing
specimens to fit the robotic setup is indispensable for achieving optimal results. After raw data
acquisition, the reconstruction of tomographic volumes poses a significant technical challenge,
being time-consuming and computationally demanding. Nevertheless, experiences from Antscan
will help drive future workflow improvements, such as the generalization and automation of
tomography pipelines or the automated post-processing of imaging data. The total storage
demands for the present datasets and raw data exceed 100 Terabytes, thereby necessitating long-
term commitment from storage-hosting institutions. Further, it is imperative to ease data
processing for comparative research, which currently represents a major bottleneck^{20,47}.

Computer vision methods, particularly neural networks, have the potential to reliably identify
tissues in micro-CT data, as shown recently for ant and bee brains^{21,22} and as we show here with
the automated generation of whole-body segmentations. When developed further, the Antscan
project design can be scaled to hundreds of thousands and potentially millions of specimens. Its
imaging principles allow both in-depth examination of individual specimens and fast screening
and automated analysis of large data subsets, which will be crucial in future research based on
digital morphology.

High-throughput synchrotron micro-CT in close collaboration with collectors and collection
managers promises a key solution for digitizing 3D anatomy across small invertebrates. A
remaining significant obstacle to the broader use of synchrotron tomography is the limited
beamtime availability at synchrotron facilities. Most of them offer user service based on peer-
reviewed proposals, and obtaining beamtime remains competitive. Automation of synchrotron
facility imaging stations through robots and the resulting high-throughput setups are also
developed to very different degrees. Moreover, the homogeneously illuminated field of view at
many synchrotron X-ray imaging beamlines is restricted, which impedes the scanning of larger
specimens. However, current developments, such as longer beamlines⁴⁸ or Bragg crystal optics⁴⁹,
and novel sample exchange systems point towards a broader range of organism and collection
sizes to be imaged with comparable synchrotron micro-CT setups.

The scientific community benefits from a rapidly growing collection of commercial and open-
access software that facilitates analysis and visualization of 3D volumetric datasets^{50,51}. Volume
renderings based on grey values can be employed to quickly generate impressive 3D views^{52,53},
while surface meshes allow digital dissections and the creation of interactive 3D models^{45,54}.
Segmentation remains the most common feature extraction method for CT scans⁵¹, and the
contrast properties across Antscan datasets facilitate this process for users. Due to the diversity
of specimens sampled, Antscan offers the possibility to visualize external and internal
morphology on larger comparative scales than previously possible with tomographic data,
especially for insects.

In contrast to physical specimens, digital information can be directly accessed from anywhere in
the world, enabling immediate and simultaneous access by researchers, artists, and the public,
thus allowing wider and more equitable engagement with biodiversity. Antscan is a scalable
approach to digitize small-bodied organisms in 3D to make the “micro” world more accessible.
Like a genome, a 3D scan contains deep information about an organism, but obtaining
quantitative information from micro-CT scans remains challenging. We will need new
bioinformatics based on automated image analysis to fully unlock the potential of databases like
Antscan, but recent developments in this area show promise^{21,22}. The Antscan initiative aims to
empower and encourage people around the world to engage with and incorporate highly resolved
ant morphology and anatomy into their science. With the approach described here and further
developments in imaging technology, bioinformatics, and artificial intelligence on the horizon, it

is time for the study of phenotypes to take its place alongside other big data endeavors in
biology.

Materials & methods

Material

We gathered ants preserved in ethanol from museum-, university-, and private collections. 132
specimens were previously stained with iodine (see Supplemental Table S2). At OIST, we
checked the vials, chose apparently well-preserved specimens, transferred them to fresh 99%
ethanol, assigned unique specimen IDs, and, depending on their size, stored ants individually in
0.2 ml, 0.5 ml, and 1.5 ml plastic reaction tubes that would fit in the robotic setup at KIT. We
extracted and databased label data from the specimens to link relevant ecological metadata to the
specimens. We then preliminarily sorted specimens into scan trays by size, each tray containing
48 vials for automated scanning. To optimize the sorting step prior to scanning, specimens within
one tray should have about the same size within the tube. That way, they can be assigned
conveniently to the different available magnifications, and if necessary, the number of necessary
height steps can be estimated for the tray. We labeled vials with a code to trace each specimen
back to its respective metadata. Tube labels were then the basis for file names during scanning.
To make this physical identifier as legible and permanent as possible, flat snap-cap plastic tubes
with ideally a machine-written code attached to the top of the lid have proven useful. All
specimens are being stored indefinitely at OIST or have been returned to their owners or
managing institutions.

High-throughput synchrotron X-ray microtomography

The specimens were scanned within two campaigns at the IPS Imaging Cluster at the KIT Light
Source. Due to the available access, two different beamlines were used. During both campaigns,
the same high-throughput tomography experimental station⁵⁵ was employed, ensuring identical
detector systems, available magnifications, scanning resolutions and sample exchange robotics
(Advanced Design Consulting USA, Inc.). This guarantees the comparability of datasets at a
given magnification setting within each measurement campaign

In the first campaign, a set of samples was investigated using a parallel polychromatic X-ray
beam produced by a 1.5 T bending magnet, spectrally filtered by 0.5 mm aluminum to remove
low energy components from the beam⁵⁶. The resulting spectrum peaked at about 17 keV, and a
full width at half maximum bandwidth of about 10 keV. In the second campaign, the samples
were scanned by using a beam diffracted by a Double Multilayer Monochromator (DMM). The
measurements were performed at a magnetic field of the wiggler of 2.7 T, yielding a maximum

flux density at 16 keV with an energy bandwidth of 2%. To reduce the heat load on the DMM,
the beam was pre-filtered with pyrolytic graphite.

Depending on their size, ants were scanned with magnifications of 10x, 5x, or 2x, resulting in
effective pixel sizes of 1.22 μm , 2.44 μm , or 6.11 μm , respectively. An air-bearing rotary stage
(RT150S, LAB Motion Systems) served for sample rotation. A fast indirect detector system
consisting of a scintillator (10x: 13 μm LSO: Tb; 5x: 25 μm LSO: Tb; 2x: 200 μm LuAG: Ce), a
double objective white beam microscope⁵⁷ (Optique Peter) and a 12-bit pco.dimax high-speed
camera (Excelitas PCO GmbH) with 2016 \times 2016 pixels was employed. For each scan, 200 dark
field images, 200 flat field images, and 3000 equiangularly spaced radiographic projections in a
range of 180° were taken with exposure times between 6.25 milliseconds and 25 milliseconds,
resulting in scan durations between 21 and 85 seconds. If the ants were too large for the field of
view, additional height steps were scanned. In total, we acquired 4010 scans (10x: 165; 5x: 3026;
2x: 807; laboratory micro-CT: 12) for 2193 individual ants plus 32 outgroup Hymenoptera (10x:
149; 5x: 1831; 2x: 239; laboratory micro-CT: 6).

[revised manuscript text omitted]

Data availability

All processed Antscan tomograms are stored at KIT's Large Scale Data Facility and integrated
into Biomedisa for public access (<https://biomedisa.info/antscan/>). Additionally, all datasets are
archived at the RADAR4KIT repository (<https://biomedisa.info/antscan,>
<https://radar.kit.edu/radar/en/search?query=antscan> *Data on the RADAR4KIT system is in a
review stage but can be directly accessed from individual links provided in Supplemental Table
S2*). Both versions of the database and all metadata linked to datasets is accessible via
<https://antscan.info> *Password for review status: antscanreview*. All metadata is also presented
in the supplemental information to this publication. Raw X-ray projections and all other image

[revised manuscript text omitted]

*B: Beam Interactions with Materials and Atoms* **267**, 1978-1988,
doi:10.1016/j.nimb.2009.04.002 (2009).
- Douissard, P. A. *et al.* A versatile indirect detector design for hard X-ray microimaging.
*Journal of Instrumentation* **7**, P09016-P09016, doi:10.1088/1748-0221/7/09/p09016
(2012).
- Vogelgesang, M. *et al.* Real-time image-content-based beamline control for smart 4D X-
ray imaging. *J Synchrotron Radiat* **23**, 1254-1263, doi:10.1107/S1600577516010195
(2016).
- Paganin, D., Mayo, S. C., Gureyev, T. E., Miller, P. R. & Wilkins, S. W. Simultaneous
phase and amplitude extraction from a single defocused image of a homogeneous
object. *J Microsc* **206**, 33-40, doi:10.1046/j.1365-2818.2002.01010.x (2002).
- Faragó, T. *et al.* syris: a flexible and efficient framework for X-ray imaging experiments
simulation. *J Synchrotron Radiat* **24**, 1283-1295, doi:10.1107/S1600577517012255
(2017).
- Faragó, T. *et al.* Tofu: a fast, versatile and user-friendly image processing toolkit for
computed tomography. *J Synchrotron Radiat* **29**, 916-927,
doi:10.1107/S160057752200282X (2022).
- Lowekamp, B. C., Chen, D. T., Ibanez, L. & Blezek, D. The Design of SimpleITK. *Front*
*Neuroinform* **7**, 45, doi:10.3389/fninf.2013.00045 (2013).
- Yaniv, Z., Lowekamp, B. C., Johnson, H. J. & Beare, R. SimpleITK Image-Analysis
Notebooks: a Collaborative Environment for Education and Reproducible Research. *J*
*Digit Imaging* **31**, 290-303, doi:10.1007/s10278-017-0037-8 (2018).
- Lösel, P. & Heuveline, V. Enhancing a diffusion algorithm for 4D image segmentation
using local information. *Proc. SPIE* **9784**, 97842L (2016).
- Ayachit, U. *The paraview guide: a parallel visualization application.* (Kitware, Inc.,
2015).
- Schindelin, J. *et al.* Fiji: an open-source platform for biological-image analysis. *Nat*
*Methods* **9**, 676-682, doi:10.1038/nmeth.2019 (2012).
- Lösel, P. D. GPU-basierte Verfahren zur Segmentierung biomedizinischer Bilddaten.
*Thesis Doktor rer. nat.*, doi:10.11588/heidok.00031525 (2022).

Coordination

Sample Preparation

High-throughput X-ray microtomography

Open Repository

high-throughput micro-CT across the ant tree of life

ant specimens | 2193
 ant species | 784
 ant genera | 210
 workers | 1674
 males | 218
 queens | 288
 species w/ genome | 183
 outgroups | 32

Myrmicinae

Ectatomminae

Formicinae

in antscan
 yes
 not yet

Leptanillinae Martialinae Amblyoponinae Proceratiinae Paraponerinae Agroecomyrmecinae

Ponerinae

Dorylinae

Myrmeciinae Pseudomyrmecinae

Dolichoderinae

Aneuretinae

μ CT, 8.4 μ m pixel size

SR- μ CT, 6.1 μ m pixel size

SR- μ CT, 2.44 μ m pixel size

SR- μ CT, 1.22 μ m pixel size

a

b

c

d

e

f